# Model-based decoupling of evoked and spontaneous neural activity in calcium imaging data

**Marcus A. Triplett[1,2], Zac Pujic[1], Biao Sun[1], Lilach Avitan[1], Geoffrey J. Goodhill[1,2]***

**1** Queensland Brain Institute, The University of Queensland, St Lucia, Australia, **2** School of Mathematics and Physics, The University of Queensland, St Lucia, Australia

* g.goodhill@uq.edu.au

**Data Availability Statement:** Code for fitting the CILVA model and data for the example zebrafish in Figs 1–4 are available at https://github.com/

## Abstract

The pattern of neural activity evoked by a stimulus can be substantially affected by ongoing spontaneous activity. Separating these two types of activity is particularly important for calcium imaging data given the slow temporal dynamics of calcium indicators. Here we present a statistical model that decouples stimulus-driven activity from low dimensional spontaneous activity in this case. The model identifies hidden factors giving rise to spontaneous activity while jointly estimating stimulus tuning properties that account for the confounding effects that these factors introduce. By applying our model to data from zebrafish optic tectum and mouse visual cortex, we obtain quantitative measurements of the extent that neurons in each case are driven by evoked activity, spontaneous activity, and their interaction. By not averaging away potentially important information encoded in spontaneous activity, this broadly applicable model brings new insight into population-level neural activity within single trials.

## Author summary

An important question in neuroscience is how the joint activity of populations of neurons encode sensory information. This can be challenging to answer because neural populations activate spontaneously, biasing stimulus-response estimates. Calcium imaging, now a dominant modality for monitoring such neural population activity, suffers especially from this effect as calcium transients are markedly slow. By simultaneously modelling the contribution of sensory stimuli and hidden sources of spontaneous activity to calcium imaging data, we demonstrate that evoked and spontaneous activity can be explicitly decoupled on a single-trial basis, leading to estimates of how neurons are relatively driven by external stimuli and latent internal factors.

## Introduction

The nervous system constructs internal representations of its sensory environment by coordinating patterns of neural activity. Uncovering these representations from neural recordings is

GoodhillLab/CILVA. Data used for Fig 5 is available at ref. [32].

**Funding:** This work was supported by Australian Research Council Discovery Projects 170102263 and 180100636 awarded to G.J.G (www.arc.gov. au). M.A.T. was supported by an Australian Government Research Training Program Scholarship. The funders had no role in study design, data collection and analysis, decision to publish, or preparation of the manuscript.

**Competing interests:** The authors have declared that no competing interests exist.

a central problem in systems neuroscience. Typically this task is approached by measuring the relationship between the parameters of a stimulus and the intensity of the neural response following stimulus presentation. However, the pattern of neural activity evoked by a stimulus is highly variable, and is usually different each time the stimulus is presented. An important source of this variability is ongoing spontaneous activity (SA) that does not appear to be driven by the stimulus [1]. In some cases this SA may simply be biophysical noise that should be averaged away, but in other cases it may represent salient features of brain function such as parallel encoding of non-sensory variables [2, 3], mechanisms for circuit development [4], or other internal-state factors that regulate sensory-guided behaviour [5]. Uncovering the interplay between stimulus-evoked activity (EA) and SA therefore requires the ability to reliably separate these two components. This is challenging, however, because the internal factors that give rise to SA are often unknown or cannot currently be directly measured.

This problem is particularly acute for calcium imaging data, a major source of our current understanding of the joint activity of large numbers of neurons. Neurons that express calcium indicators report activity at a high spatial resolution, but filter out high frequency spiking due to slow indicator binding kinetics and saturating calcium concentrations [6, 7]. These calcium levels in turn are only observed through temporally subsampled fluorescence intensities that are subject to noise from the optical imaging system. Moreover, neurons can be recorded in large populations with many thousands of imaging frames, leading to very high dimensional data that can challenge traditional methods of neural data analysis [8].

Much research in recent years has focused on statistical methods for extracting hidden (or "latent") structure from neural population data [9–12]. A key assumption in these methods is that neural population activity tends to possess a characteristic low dimensional structure, reflecting underlying constraints on how neurons can comodulate their activity [13]. Thus high dimensional neural data can often be well-described by a much smaller number of latent variables evolving through time. In this context, unobserved sources of SA are latent variables that can be inferred from data given the appropriate statistical tools. However, methods for identifying latent structure in calcium imaging data (see e.g. refs. [14–16]) are scarce compared to spike train data, and none so far have sought to explicitly extract sources of SA hidden amidst population responses to sensory stimuli.

Here we develop a latent variable model for calcium imaging data that allows for a decomposition of single-trial neural activity into its evoked and spontaneous components. In our model, which we refer to as calcium imaging latent variable analysis (CILVA), patterns of SA are driven by hidden factors decoupled from the stimulus. By fitting the model to data, we identify the structure and temporal behaviour of these latent sources of SA, and simultaneously extract receptive fields that are not biased by the variability that these sources of SA introduce. Many analyses of calcium imaging data deconvolve calcium transients to estimate the underlying neural activity before using more traditional methods of spike train analysis. Here we jointly model the underlying sources of activity together with the calcium transients themselves, allowing a direct comparison between the raw imaging data and the model components, and avoiding the intermediate computational step of deconvolution, which can impact model performance compared to joint inference approaches (see e.g. [14]).

To demonstrate the applicability of the model we analysed calcium imaging data from both the larval zebrafish optic tectum and mouse visual cortex. In both cases we identified sparsely active independent latent factors that targeted distinct sets of neurons. Besides revealing the statistical structure of SA, accounting for these factors produced sharper receptive field estimates, more refined retinotopic maps, and quantitative measurements of the presence and interaction of EA and SA. Together, these results show that CILVA is an effective new approach for single-trial analysis of calcium imaging data.

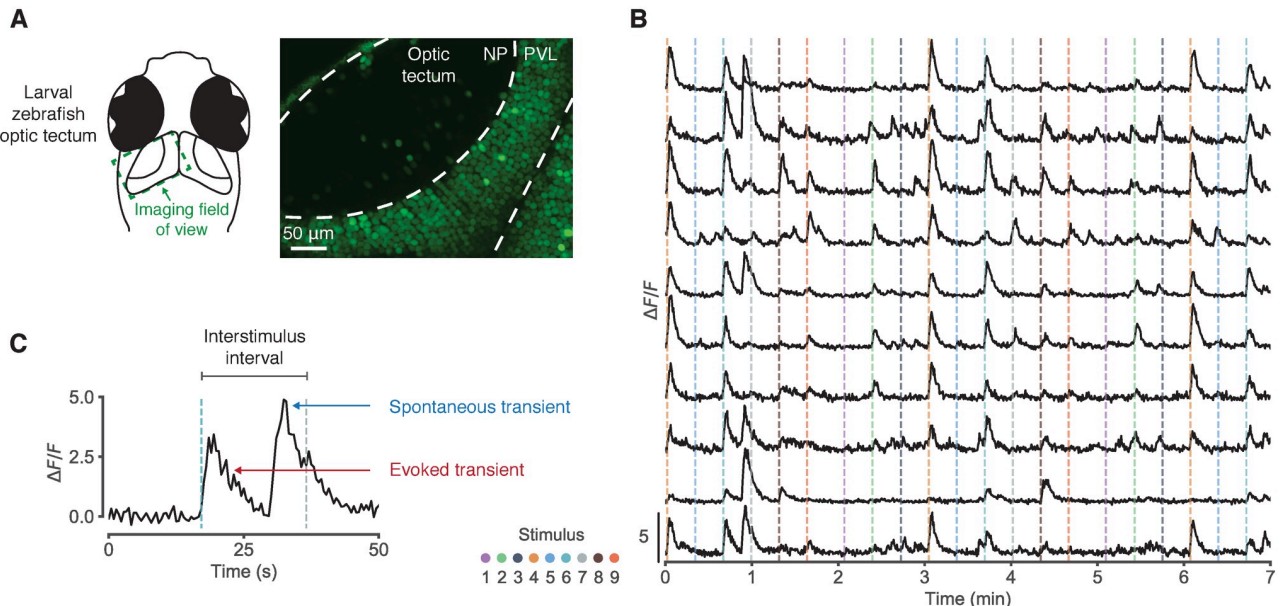

**Fig 1. Spontaneous activity in calcium imaging data.** (A) Two-photon calcium imaging of the larval zebrafish optic tectum. NP, neuropil; PVL, periventricular layer. (B) Fluorescence traces from 10 example neurons. Dashed vertical lines indicate stimulus onset; colour represents azimuth angle of presented stimulus. (C) Example fluorescence trace segment illustrating that spontaneous calcium transients can occur just before stimulus onset, inflating stimulus-response estimates.

## Results

### Low dimensional spontaneous activity proceeds throughout stimulus presentation

We first considered two-photon calcium imaging data from the optic tectum of the developing larval zebrafish (Fig 1A). Fish expressing the genetically encoded calcium indicator GCaMP6s were embedded in agarose while small dark spots were presented at systematically varying angles across the visual field [17]. Onset of the visual stimulus evoked calcium transients in the tectum (Fig 1B) consistent with the topography of the retinotectal map. The presentation of a spot was followed by an interval of 19s without any stimulation, enough time for calcium levels to return to baseline before the next stimulus. The optic tectum was often highly active during these inter-stimulus intervals (Fig 1C) and calcium transients sometimes occurred spontaneously just before stimulus onset, elevating the recorded fluorescence levels associated with that stimulus.

One approach to separating SA from EA would be to repeatedly present the same set of visual stimuli over many trials, compute the peri-stimulus time histogram (PSTH) across trials, and assign all calcium transients that deviate from the PSTH as "spontaneous". However, this approach cannot handle stimulus sequences that occur within single trials or in a randomised order unless the PSTH is calculated over a short window surrounding the time of stimulus onset, in which case the estimated SA is biased by edge effects. To the authors' knowledge there are no standard existing methods capable of separating SA from EA on a single-trial basis.

As our first attempt we therefore considered using a novel combination of existing tools. We characterised the stimulus-driven component of the population activity by simply expressing each fluorescence trace as a linear combination of regressors that specify the basic shape and timescale of calcium activity. We defined a basis of stimulus regressors [18, 19] by convolving a calcium impulse response kernel with the presentation times of each stimulus

and performed a multivariate linear regression of the population data onto this basis set using non-negative least squares (S1A–S1C Fig). Any highly structured activity in the residual data would likely be driven primarily by latent sources of SA not directly related to the onset of the presented stimuli. We then applied non-negative matrix factorisation [20] (NMF) to search for low dimensional structure in the residuals. NMF attempted to reconstruct the residual population data as the product of two matrices with non-negative entries: a matrix whose rows are timeseries that capture patterns of SA shared across groups of neurons, and a matrix whose columns describe how neurons are coupled to such timeseries (S1D Fig). The NMF description of SA identified low dimensional structure that proceeded throughout the recording, largely independent of the stimulus (S1E Fig).

While this residual NMF approach is efficient due to highly optimised computational routines, it is limited by two characteristics. First, because the models of stimulus processing and shared SA are not inferred jointly, receptive field estimates are biased towards higher values by spontaneous calcium transients that coincide with stimulus presentations. This in turn introduces bias when applying NMF to the residual data, since some of the contribution of SA was already subtracted out at the receptive field estimation stage. Second, NMF has no model for the highly stereotyped structure of calcium transients, and therefore does not respect this structure in the components that it finds (S1F Fig).

## CILVA simultaneously captures evoked responses and shared spontaneous activity

To overcome these difficulties, we instead developed a generative statistical model that describes evoked and spontaneous activity simultaneously rather than sequentially (Fig 2A and 2B). Our method works directly with filtered fluorescence traces of individual neurons rather than raw calcium imaging videos, and therefore can be used after segmenting cells with popular calcium imaging preprocessing packages such as CaImAn [21] and Suite2p [22]. While these packages provide spike deconvolution modules, we opted to work with fluorescence traces as many calcium imaging datasets (including those analysed in this paper) have low temporal resolution, rendering precise estimation of spike times difficult. With this in mind, the model specifies the observed fluorescence level $f_n(t)$ for neuron $n$ at time $t$ in terms of the underlying calcium concentration $c_n(t)$,

$$f_n(t) = \alpha_n c_n(t) + \beta_n + \epsilon_n(t)$$

where the scalars $\alpha_n$ and $\beta_n$ determine the scale and baseline of the fluorescence signal respectively, and $\epsilon_n(t)$ represents Gaussian noise. Consistent with experimental data [6] and previous models for calcium imaging [7, 23], the calcium dynamics are assumed to be highly stereotyped and are defined by the convolution of a GCaMP impulse response kernel $\mathbf{k}$ with a vector of calcium influxes $\boldsymbol{\lambda}_n$ (analogous to an activity intensity function).

$$c_n(t) = \sum_{\tau=0}^{t} k(t - \tau)\lambda_n(\tau).$$

The kernel $\mathbf{k}$ is a difference-of-exponentials function (see Methods), which includes both rise and decay time constants. We found that including an explicit rise time was essential as GCaMP6s activity is slow relative to the sampling rate of many optical imaging systems.

The key ideas of the model are that (i) evoked responses will tend to be locked to the onset of the stimulus, (ii) evoked responses typically have a simple impulse response structure in calcium imaging data, and (iii) neural activity not attributable to evoked activity should be explained as far as possible by SA with a specific structure. The rate of calcium influx $\lambda_n(t)$ in

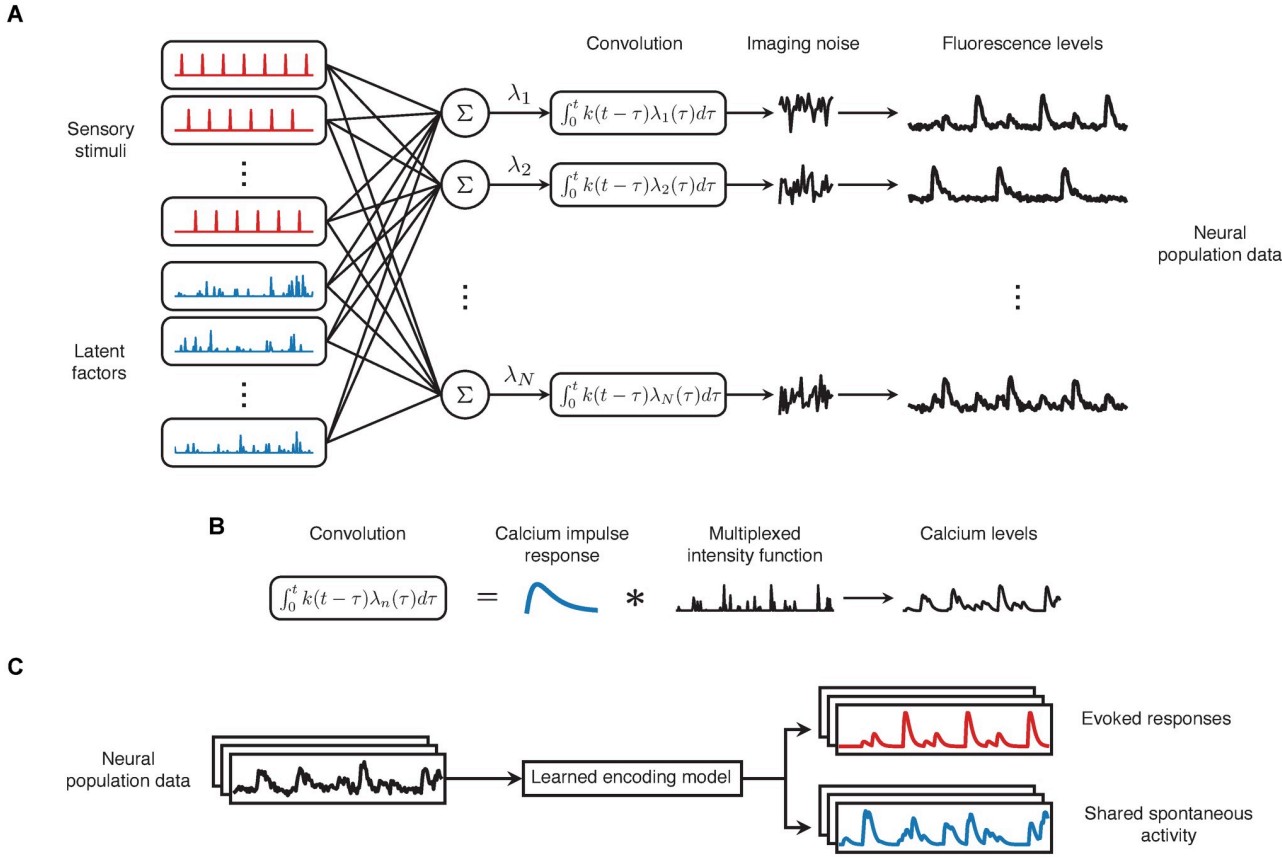

**Fig 2. Overview of the CILVA approach for decoupling stimulus-evoked responses and latent sources of SA.** (A) Proposed generative architecture underlying multivariate calcium imaging data. Neurons are driven by sensory stimuli (red) and latent sources of SA (blue). These two sources are combined additively to define the underlying rate of calcium influx ($\lambda_n$), before being convolved with a GCaMP kernel. Calcium levels are subsequently reported through noisy fluorescence intensities. (B) The intensity of calcium influx $\lambda_n$ encoding stimuli and shared SA is convolved with a GCaMP kernel **k** to generate observed calcium levels. (C) The learned encoding model provides a method for decoupling evoked responses from common patterns of SA.

each imaging frame $t$ was thus assumed to be driven by the addition of two underlying non-negative sources: processing of the stimulus $\mathbf{s}(t)$ through a linear receptive field $\mathbf{w}_n$, and a small number of unobserved or "latent" sources of SA $\mathbf{x}(t)$,

$$\lambda_n(t) = \mathbf{w}_n^\top \mathbf{s}(t) + \mathbf{b}_n^\top \mathbf{x}(t).$$

Here $\mathbf{x}(t)$ is the low dimensional latent state at time $t$, $\mathbf{b}_n$ is a vector describing how neuron $n$ is affected by these factors, and $\cdot^\top$ denotes the transpose operation. In the event that neurons exhibit prolonged neural responses, the stimulus design matrix **s** can be straightforwardly modified by including copies of each stimulus shifted in time [24]. This SA model is inspired by the application of factor analysis methods to neural population data [9, 25], which posit that low dimensional structure arises from latent computations or brain states that concurrently affect subsets of neurons. However, the latent factors underlying spontaneous calcium transients in our model differ mathematically from classical factor analysis in that, due to non-negativity of the calcium levels, factor activity states and coupling between latent factors and neurons must be non-negative [26].

We fit the model by computing the maximum *a posteriori* estimate of the latent factor activity states. Because these activities were less constrained by the model compared to the time-locked evoked responses (and therefore likely to be more complex) we encouraged sparsity by placing a non-negative prior on the latent factors with high density near zero, and used a simple model selection procedure to estimate the sparsity penalty (see Methods).

## Fluorescence signals can be decomposed into their evoked and spontaneous components

Our model can be used to analyse the separate contributions of evoked and spontaneous activity to the observed fluorescence levels (Fig 2C). The fitted model can be succinctly summarised by the equation

$$\hat{\mathbf{f}}_n = \hat{\alpha}_n \mathbf{k} * (\hat{\mathbf{w}}_n^\top \mathbf{s} + \hat{\mathbf{b}}_n^\top \hat{\mathbf{x}}) + \hat{\beta}_n \mathbf{1}_T$$

where $\hat{\ }$ denotes an estimated variable, $*$ denotes linear convolution, and $\mathbf{1}_T$ is a vector of ones with length equal to the number of imaging frames $T$. Here we have dropped explicit dependence on the calcium levels $c_n(t)$, which are deterministic given the other model parameters. The components of the signal driven purely by evoked or spontaneous activity can then be extracted from the convolution to give

$$\hat{\mathbf{f}}_n^{\text{evoked}} = \hat{\alpha}_n \mathbf{k} * \hat{\mathbf{w}}_n^\top \mathbf{s} + \hat{\beta}_n \mathbf{1}_T$$

and

$$\hat{\mathbf{f}}_n^{\text{spont}} = \hat{\alpha}_n \mathbf{k} * \hat{\mathbf{b}}_n^\top \hat{\mathbf{x}} + \hat{\beta}_n \mathbf{1}_T.$$

We first verified the model on simulated data with known ground truth, modelling the properties of the zebrafish and mouse data that we subsequently consider (S1 and S2 Tables, S2 and S3 Figs). We then applied the model to our calcium imaging data from the zebrafish optic tectum to decouple the evoked and spontaneous calcium transients (Fig 3). These data were segmented using custom software, but we also verified that our results do not depend on the preprocessing package for source extraction by performing the same analysis on data processed with CaImAn (S4 Fig). Overlaying these decoupled calcium traces onto the experimental data, we found that they provided realistic descriptions of calcium activity (Fig 3A) and a close fit between the raw fluorescence trace and the model reconstruction (Fig 3B, S5 Fig). The time-locked responses to stimuli were well modelled by $\hat{\mathbf{f}}_n^{\text{evoked}}$, while low dimensional SA was identified by the projected latent factor activity $\hat{\mathbf{f}}_n^{\text{spont}}$ (Fig 3A, S6 Fig, S1 and S2 Videos). Neural activity in the residual data (i.e., after subtracting the model reconstruction from the raw fluorescence traces) arose primarily from spontaneous calcium transients that were independent of the latent sources of shared SA (and were therefore attributed to private, as opposed to shared, variability [27], S7 Fig).

We fit the model with three latent factors, whose inferred activity timeseries were sparse (Fig 3C). Including additional latent factors beyond these resulted in better models of the SA of individual neurons or small subsets of neurons, but caused little improvement in the overall quality of model fit for this fish (Fig 3D). To understand the relative importance of each factor to the overall model fit, we defined a contribution index for a factor as the average reduction in the quality of model fit following its deletion (Methods). We found that each factor made a substantial contribution to the overall model fit by modulating shared SA across large groups of tectal neurons (Fig 3E and 3F).

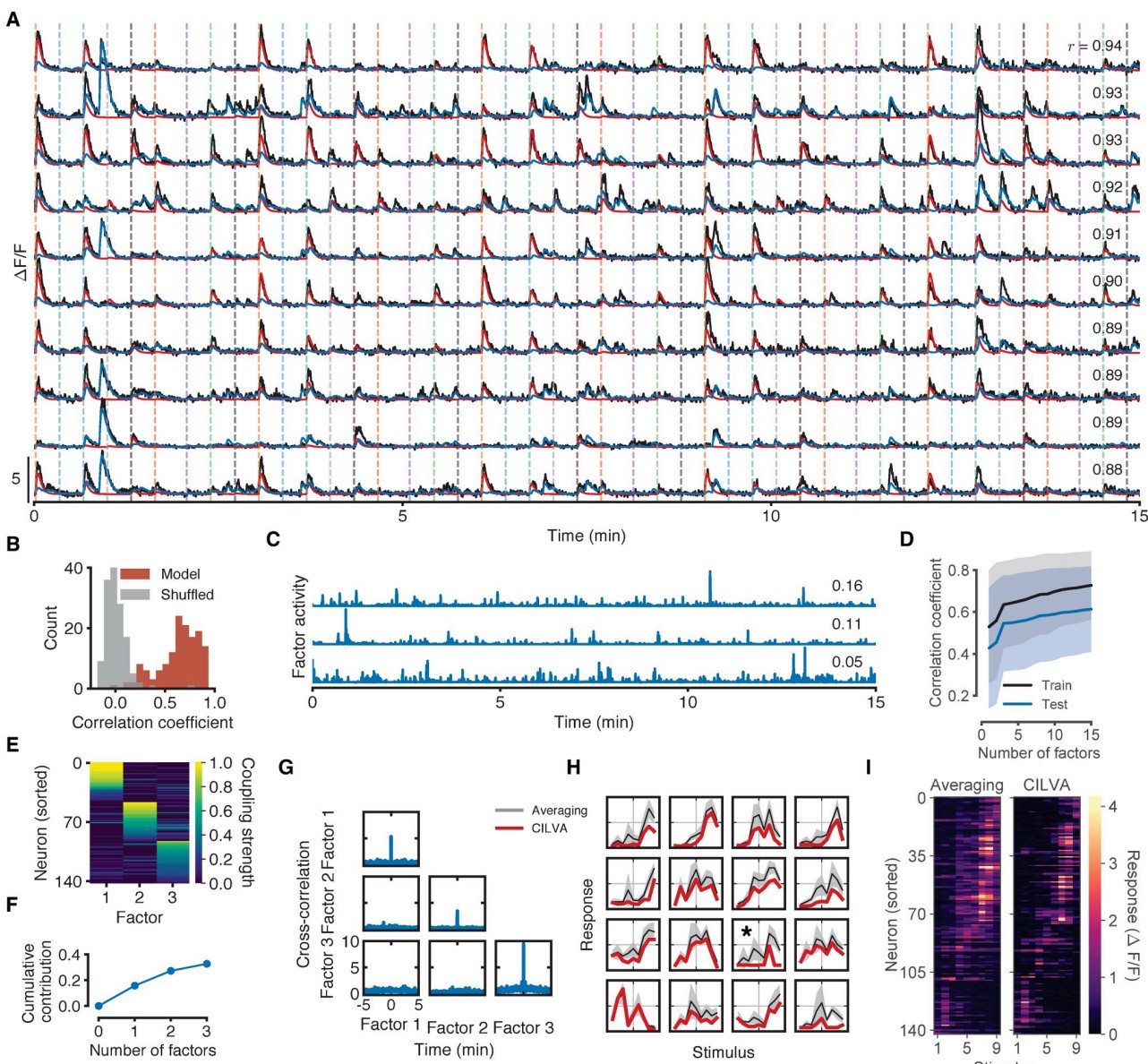

**Fig 3. Fitted model components for the zebrafish shown in Fig 1.** (A) Results of fitting CILVA and decoupling EA (red) and shared SA (blue) in an experimental recording. Inset numbers denote the Pearson correlation coefficient between raw fluorescence trace and model fit. The 10 neurons with the highest correlations between data and model fit are shown. (B) Distribution of correlation coefficients between data and model fits. Shuffled data obtained by cyclically permuting each trace by a random offset while preserving its temporal structure. (C) Inferred latent factor timeseries. Inset numbers denote the factor contribution indices, defined as the mean reduction in correlation coefficient across the population following deletion of the corresponding factor. (D) Cross-validated distributions of correlation coefficients for 1 to 15 latent factors. Shaded error bars indicate one standard deviation. For this fish adding additional latent sources of SA beyond three factors provides little improvement in the average correlation for both training and held-out test data. (E) Estimated factor coupling matrix shows that latent factors target distinct, non-overlapping sets of neurons. (F) Cumulative factor contribution indices for 0 to 3 latent factors. (G) Cross-correlograms show little interaction between latent factors and no long term structure in individual factor activity. While zebrafish can occasionally exhibit factors with secondary peaks in their autocorrelation plots (see e.g. S9H Fig), the location of these peaks varies between factors and fish, and do not align with the interstimulus interval. (H) Example estimated stimulus tuning curves (red). Tuning curves obtained by trial-averaging fluorescence levels over a window following stimulus onset (gray) provided for comparison (Methods). Shaded error bars denote one standard deviation. Full temporal traces for these neurons are given in S8 Fig. (I) Retinotopic maps obtained by trial-averaging fluorescence levels over a window following stimulus onset (left) and by CILVA-estimated tuning curves (right). The CILVA map is more refined since the estimated stimulus filters already account for ongoing SA.

The factor coupling matrix (defined by the vectors $\mathbf{b}_n$) reports how neurons are affected by the latent factors. There are several possibilities for how this matrix could be structured. First, if there is a minimal presence of structured SA the coupling matrix may exhibit no coherent organisation at all. Second, neurons could require the coordinated activity of several latent factors to explain their SA. This would be the case if, e.g., neurons participated in multiple recurrently connected circuits driven by noise [28, 29], and would result in factors modulating overlapping groups of neurons. Third, latent factors may each drive their own distinct sets of neurons, with little cross-talk between them. This could occur if, e.g., latent factors were encoding unrelated streams of motor or non-visual sensory information [2, 30]. In our example zebrafish the estimated coupling matrix had a highly modular structure, with factors influencing largely non-overlapping sets of neurons (Fig 3E). Furthermore, the factor cross-correlograms showed no sign of dependence between factors, indicating that distinct sets of neurons were uniquely targeted by independent latent sources of SA (Fig 3G).

Since our model fits receptive fields jointly with latent sources of SA, the estimated tuning curve for each neuron already accounts for ongoing SA that may have inflated its responses to stimuli. Indeed, if spontaneous calcium transients coincide with the presentation of a stimulus, one could expect tuning curves obtained by simply averaging the fluorescence levels over a small window following stimulus onset to be spuriously larger, and exhibit higher variance than if these events did not occur. We plotted the tuning curves estimated by CILVA against tuning curves obtained by averaging (see Methods) and found that they confirmed this intuition (Fig 3H). Moreover, sorting the neurons according to their preferred stimulus revealed a more refined retinotopic map when explicitly accounting for SA (Fig 3I). While our data showed a variety of tuning types, of note were neurons that were unselective to visual stimuli (i.e., had relatively flat tuning curves), but that were highly active throughout the recording (e.g., the neuron marked by an asterisk in Fig 3H and S8 Fig).

## Neurons are differentially driven by external stimuli and latent internal factors

To quantify the extent to which each neuron is driven by sensory stimuli versus shared SA, we derived an equation that expressed the variance of the reconstructed fluorescence levels in terms of three components: the variance attributable solely to EA, the variance attributable solely to shared SA, and the covariance (i.e., interaction) between EA and shared SA (Fig 4A, Methods). This revealed that across the population there was a continuous progression of responses, with some neurons being primarily driven by EA, some primarily by SA, and some by a mixture of both SA and EA (Fig 4A). To confirm that these effects were not artefacts of the model or the calcium indicator, we verified that the model does not overestimate the variance in the data (Fig 4B) and that there were interactions between EA and SA that were greater than expected by chance (Fig 4C). We defined a "drive ratio" to measure whether neurons were driven more by SA or EA (Methods). The distribution of drive ratios was largely bimodal (Fig 4D), indicating a preference to be dominated by either EA or SA rather than responding equally to both.

We next quantified the improvement that resulted from incorporating SA into the model. Without SA, the model for each neuron consists of a simple linear filter convolved with a calcium kernel. This is a good description of neurons that possess high drive ratios (i.e., whose variances are dominated by EA) and thus these neurons show little improvement in how well the statistical model fits their fluorescence levels with the incorporation of SA (Fig 4E, neurons along the diagonal). In contrast, many neurons are poorly fit by a model that incorporates only stimulus responses, and show substantial improvement when shared sources of SA are

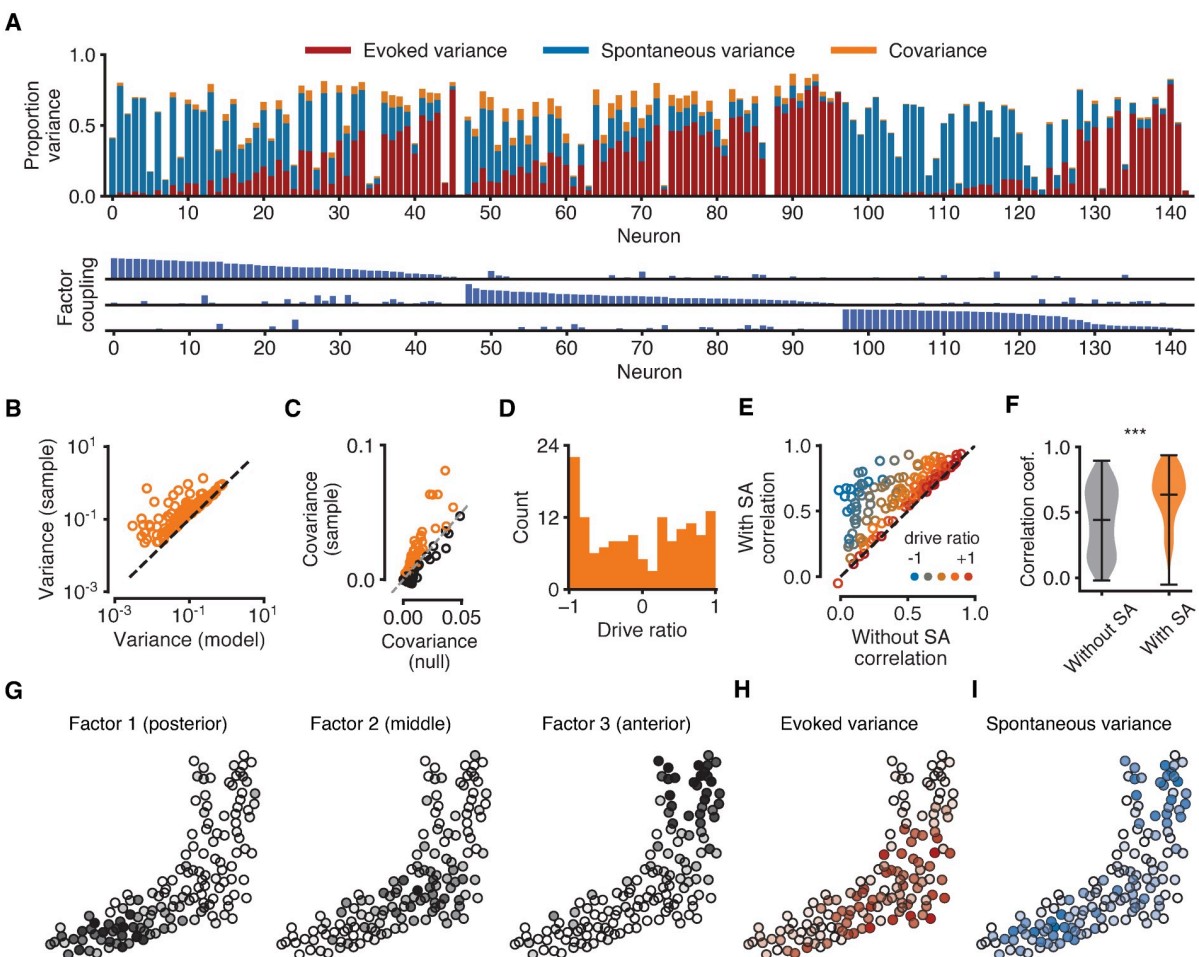

**Fig 4. Analysis of the contribution of EA and SA to neural variability.** All data for the same fish as in Fig 3. (A) Top: composition of each neuron's sample variance in terms of variance attributable solely to EA (red bars), solely to shared SA (blue bars), and their covariance (orange bars). Orange bars represent absolute values of covariances for ease of visualisation. Variance components are given as proportions of the total sample variance of the raw fluorescence signal $\text{var}[\mathbf{f}_n]$ (corrected for imaging noise, see Methods). Neurons sorted by the strength of their coupling to each factor (as in Fig 3E). Bottom: coupling between neurons and latent sources of SA suggests neurons with strong coupling are weakly driven by sensory stimuli. Maximum bar height of one. (B) Sample variance (corrected for imaging noise) vs variance of the statistical model indicates that the model does not overestimate variance. Each data point represents one neuron. (C) Covariances between evoked and spontaneous traces estimated by the model (vertical axis). Chance levels for a null model (horizontal axis) are 95th percentiles of shuffled data obtained by cyclically permuting evoked traces by random offsets 1000 times while preserving temporal structure. Sample covariances exceeding chance levels (orange circles above dashed identity line) cannot be attributed to the slow timescale of the calcium indicator. (D) Distribution of drive ratios across the population of neurons. (E) Correlation coefficient between raw fluorescence trace and evoked component of model fit (without SA) and full model fit (with SA). Neurons with strongly negative drive ratios show marked improvement in the quality of model fit. (F) Violin plots showing statistically significant improvement in the average correlation coefficient between experimental data and model fits after incorporating latent sources of SA ($p < 0.001$, Wilcoxon signed-rank test). (G) Spatial organisation of latent factors underlying SA. The three non-overlapping factors are spatially localised and tile the imaging plane. (H) Spatial organisation of the evoked variance components. Cell opacity is proportional to the fraction of variance attributable to EA for the given neuron. Neurons strongly driven by EA cluster in the middle region of the tectum. (I) Same as H, but for SA. Neurons strongly driven by SA cluster in the anterior and posterior tectum.

taken into account (Fig 4E, neurons above the diagonal), leading to a significant increase in the average correlation between fluorescence traces and model fits (Fig 4F).

In the absence of sensory stimulation, SA in the optic tectum has previously been shown to exhibit a characteristic localised spatial structure [31]. We thus sought to determine whether this effect persisted when the tectum was being actively driven by sensory stimulation. The

factors underlying SA identified by CILVA concentrated in the posterior, middle, and anterior regions of the tectum, together tiling the two dimensional imaging plane (Fig 4G). Interestingly, the evoked variance component was largely confined to the middle tectum, where coupling to latent factors was weakest (Fig 4H). Conversely, the spontaneous variance component was most strongly represented at the posterior and anterior ends of the tectum, where coupling to latent factors was strongest, with little SA in the middle tectum (Fig 4I). This spatial localisation was not imposed on the data by our model, and thus is a useful post-hoc verification that the SA our model identifies is likely to be biologically salient.

To determine how representative our results were, we fit the model to a dataset of seven additional zebrafish larvae. Example fits for two of these zebrafish are given in S9 and S10 Figs. For consistency of comparison between zebrafish we again fit the statistical model with three latent factors. Across the 8 fish the mean correlation coefficient was centered at $\sim$ 0.6 (S11A Fig), with latent factors having average individual contribution indices of 0.1 (S11B Fig). Factors were also mostly non-overlapping, with only a small fraction of neurons participating in multiple factors (S11C Fig). Incorporating all three factors increased the mean correlation coefficient between raw fluorescence data and model-fit by 34% on average compared to model-fits without the SA component (S11D Fig). Finally, we found that while EA and SA tended to be balanced at the population level, individual neurons mostly biased their activity towards being either stimulus-driven or spontaneous (S11E and S11F Fig). These results show that the basic statistical properties of the data are consistent across a set of different animals.

## CILVA identifies low dimensional patterns of SA in visual cortex

We next explored the application of the model to publicly available data from mouse primary visual cortex [32]. In this case, stimuli of higher dimension were presented more rapidly than in our previous application. Briefly, head-fixed mice expressing the calcium indicator GCaMP6s (via viral injection) stood on an air-suspended ball while drifting gratings were presented across the visual field with 1 to 3 second intervals and at 8 orientations, 3 spatial frequencies, and 4 temporal frequencies (Fig 5A). We verified with simulated data that the model could accurately recover evoked and spontaneous components in this regime (S3 Fig, S2 Table), and then applied CILVA to decouple the evoked and spontaneous fluorescence components (Fig 5B–5D).

CILVA was able to extract low dimensional patterns of SA (Fig 5D, vertical bands of activity) that were much harder to discern in the raw data (Fig 5B). This included a spontaneous event that appeared to be triggered by stimulus onset (Fig 5D, first vertical band of activity) but that did not reoccur with subsequent stimulus presentations. The model reconstruction provided a good fit, with correlation coefficients much larger than in the case of shuffled data (Fig 5E). Similar to the zebrafish data, extracted latent factors were mutually independent (S12 Fig), targeted largely non-overlapping sets of neurons (Fig 5F), and were sparsely active (S12 Fig). CILVA is thus effective for discovering novel, interpretable patterns of neural activity in high dimensional cortical imaging data.

## Discussion

Neural activity elicited in response to a stimulus can be substantially affected by ongoing SA. The CILVA approach for decoupling these influences has the advantage over simpler approaches, such as the sequential application of non-negative least squares and NMF (S1 Fig), since receptive fields are inferred simultaneously with latent factors, preventing the latter from confounding measurements of the stimulus-evoked response. Not only does this allow us to estimate tuning curves that are unbiased by spontaneous calcium transients, but also to

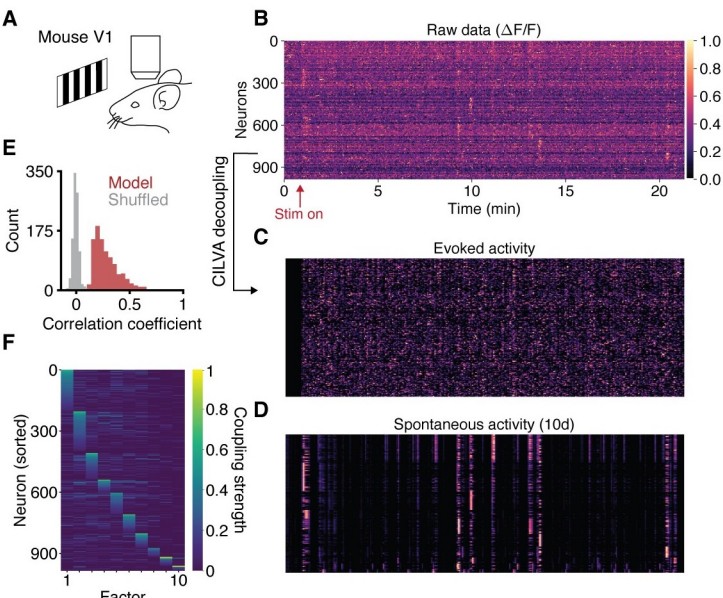

**Fig 5. Single-trial decoupling of EA and SA in visual cortex.** (A) Calcium imaging of mouse V1 during presentation of drifting gratings. (B) Raw data consists of 21 minutes of neural activity from 986 neurons. The fluorescence trace of each neuron is normalised to take values between 0 (dark) and 1 (light). Neurons sorted as in panel F. (C) Decoupled evoked component of neural activity. (D) Decoupled spontaneous component of neural activity, given a latent dimensionality of 10. (E) Distribution of correlation coefficients between data and model fits. Shuffled data obtained by cyclically permuting each trace by a random offset while preserving its temporal structure. (F) Learned factor coupling matrix showing that the inferred factors target largely non-overlapping sets of neurons.

estimate the latent structure of SA alone, unbiased by evoked responses. The composition of a neuron's sample variance can then be straightforwardly expressed in the model in terms of the variance of the decoupled evoked and spontaneous components, together with their covariance. CILVA thus provides a new tool for quantitative analyses of the interaction between EA and SA in single trials, reducing dependence on approaches to sensory coding that require averaging away potentially important information encoded in SA.

CILVA is closely related to latent factor models for spike train data. Gaussian process factor analysis, for example, assumes that population spiking activity is linearly driven by a small number of latent factors evolving smoothly through time according to a Gaussian process [9]. A similar model, the Poisson linear dynamical system, models neural activity by Poisson processes, where firing rates across the population are driven by a hidden low dimensional linear dynamical system [10]. These models consider a neuron to be a noisy sensor of an underlying latent state, and the smooth path that population activity traces through this low dimensional state space constitutes the underlying computation implemented by a neural circuit [33]. In contrast to such models, which explicitly constrain the temporal evolution of latent factors, our statistical model assumes that latent factor activity states at each time point are independent and identically distributed according to a (non-negative) maximum entropy prior. Autocorrelation of the latent factors then arises due to their convolution with a calcium impulse response function. While an explicit dynamics could be imposed on the latent factors [14], we chose not to do so due to a conflict of timescales: the relevant neural dynamics often takes place over several hundred milliseconds [9], but this may only constitute a few imaging frames in calcium imaging data. Thus, calcium transients predicted by the model may appear erroneously prolonged if factor activity states could only change gradually.

Additive interactions between EA and SA, as assumed by the model, have been identified in numerous studies. For example, optical imaging of cat visual cortical neurons using voltage-sensitive dyes [1], and cellular-resolution two-photon calcium imaging [2], multiple simultaneous Neuropixels probes [2], and wide-field calcium imaging of both cortical hemispheres [34] in mouse visual cortex have all shown substantial additive modulation of evoked responses by coordinated SA that proceeds unimpeded by stimulus onset. However, there may be cases where the interaction between EA and SA is more complex than a simple additive scheme. For example, trial-to-trial variability of evoked responses could result from changes in excitability, reflecting a multiplicative effect of SA. Such an interaction could potentially be included in our model by incorporating an appropriate nonlinear activation function (similar to ref. [35]).

The interaction between EA and SA could also affect the underlying dynamics of the neural activity. This could occur if, e.g., the presentation of a stimulus engages recurrent circuits that trigger the activation of a latent factor. In the case of our data in Fig 4H and 4I, the distinct spatial organisation of the evoked and spontaneous variance components indicate that this triggering effect is not likely to be a predominant source of variation (although there is some overlap in these two components in the middle tectum). Although this kind of "triggering" interaction is not something the model attempts to explicitly describe, CILVA can potentially account for this effect depending on how the triggering occurs. If the triggering of a factor always occurs with stimulus presentation, then this will be incorporated into the receptive field component. If the triggering of a factor occurs only occasionally and with a sufficiently large amplitude, then this will be associated with a latent factor instead.

We modelled SA as primarily originating from shared sources, with the SA of the remaining neurons arising either from private sources or from residual imaging noise. This shared SA was responsible for a substantial portion of the variance across the population, and the model accounting for shared SA significantly outperformed the model that did not (Fig 4F). However, our estimates represent merely a lower bound on the variance attributable to shared SA, and this bound could increase as the recorded set of neurons approaches the complete population. Indeed, SA that we currently consider private may in reality be shared, but due to constraints in the optical imaging system we may simply have not observed the neurons with similar profiles of SA. Okun et al. [36] found that the correlation structure of cortical populations in primates and mice could be well-predicted by the coupling of individual neurons to the population firing rate, a one dimensional measure of activity. 'Choristers' have firing rates coupled to the population, and are thus dominated by shared variability, whereas 'soloists' are less affected by population-wide events and are dominated by private variability, even during SA. In our analysis, by contrast, population activity was best described by coupling of neurons to one of multiple latent factors, and these factors could not be described by a single latent state governing SA since they were mutually independent.

Our method identified multiple independent latent sources of SA targeting distinct, largely non-overlapping sets of neurons. Even a primary sensory area like the optic tectum or visual cortex receives converging inputs from other brain regions that can make it highly active in the absence of sensory inputs [30, 37, 38]. Recently, several studies reported brain-wide activity correlated with behaviour [2, 5, 39]. For example, Stringer et al. [2] analysed calcium imaging data from 10, 000 neurons in the mouse primary visual cortex and found that locomotor variables such as pupil diameter and running speed accounted for $\sim$ 20% of the total variance of the population activity. Potentially, similar inputs to the tectum for the purpose of, e.g., visuomotor integration [18], could form the physiological basis of the latent factors that we extracted. However, while overt behavioural parameters like pupil diameter can be unambiguously measured and correlated with neural activity, CILVA attempts to adapt to any kind of

input that induces structured patterns of SA that linearly combine with stimulus-evoked responses, even if such inputs are not directly measured.

## Materials and methods

### Zebrafish recordings

All procedures were performed with approval from The University of Queensland Animal Ethics Committee (approval certificate number QBI/152/16/ARC). *Nacre* zebrafish (*Danio rerio*) embryos expressing *elavl3*:H2B-GCaMP6s, of either sex, were collected and raised according to established procedures [40] and kept under a 14/10 hr on/off light cycle.

Zebrafish larvae were embedded in 2.5% low-melting point agarose, positioned at the centre of a 35 mm diameter plastic petri dish and overlaid with E3 embryo medium. Calcium imaging was performed at a depth of 70 $\mu$m from the dorsal surface of the tectal midline. Time-lapse two-photon images were acquired using a Zeiss LSM 710 inverted two-photon microscope. A custom-made inverter tube composed of a pair of beam-steering mirrors and two identical 60 mm focal length lenses arranged in a 4f configuration was used to allow imaging with a 40X/1.0 NA water-dipping objective (Zeiss) in an upright configuration. Samples were excited via a Spectra-Physics Mai TaiDeepSee Ti:Sapphire laser (Spectra-Physics) at an excitation wavelength of 940 nm and the emitted light was bandpass filtered (500–550 nm). Laser power at the sample ranged between 12 to 20 mW. Images of 416x300 pixels were obtained at 2.1646 Hz. To improve the stability of the recording, chambers were allowed to settle for three hours prior to start of two-photon imaging.

Visual stimuli were projected on white paper placed around the wall of a 35 mm diameter petri dish using a projector (PK320 Optoma, USA), covering a horizontal field of view of 174˚. A red filter (Zeiss LP590 filter) was placed in the front of the projector to avoid interference of the projected image in the signal collected by the detector. Larvae were aligned with one eye facing the white paper side of the dish and with the body axis orthogonal to the projector. Visual stimuli were generated using custom software based on MATLAB (MathWorks) and Psychophysics Toolbox. Each trial consisted of 6˚ diameter black spots at nine different positions, separated by 15˚ intervals from 45˚ to 165˚, where 0˚ was defined as the direction of the larva's body axis. Their order was set to maximise spatial separation within a trial (45˚, 120˚, 60˚, 135˚, 75˚, 150˚, 90˚, 165˚, 105˚). Spots were presented for 1 s, followed by 19 s of blank screen. We projected consecutive trials of nine spots with 25 s of inter-trial interval.

The cell segmentation procedure is described in ref. [41]. Briefly, custom MATLAB software was used to automatically detect the region-of-interest (ROI) of each active cell, i.e., the group of pixels defining each cell. The software searched for active pixels, i.e., pixels that showed changes in brightness across frames, resulting in an activity heatmap of all the active regions across frames. The activity map was then segmented into regions using a watershed algorithm, with a similar threshold applied to all movies. Within each segmented region, we computed correlation coefficients of all pixels in the region with the mean of the most active pixel and its eight neighbouring pixels. Correlation coefficients showed a bimodal distribution; one peak of highly correlated pixels representing pixels of the cell within the region, and a second peak of relatively low correlation coefficients representing nearby pixels within the region which were not part of the cell. Using a Gaussian mixture model, we found the threshold correlation which differentiated between pixels likely to form the active cell and neighbouring pixels that were not part of the cell. We also required that each detected active area covered at least 26 pixels (5.5 mm$^2$). The software allowed visual inspection and modification of the parameter values where needed. All pixels assigned to a given cell were averaged to give a raw fluorescence trace over time.

## Mouse recordings

We used publicly available data [32]. The experimental procedures are described in detail in refs. [42, 43]. Neurons were recorded simultaneously at 2.5Hz using calcium imaging and segmented using Suite2p. Visual stimuli were shown at approximately 1 Hz, with randomized inter-stimulus intervals. The stimuli were drifting gratings with 8 directions, 4 spatial frequencies and 3 temporal frequencies. Blank stimuli (gray screen) were also interleaved. While this data was originally recorded from multiple imaging planes, to avoid issues with timing of latent factor activity between planes we restricted our analysis to neurons from a single imaging plane, resulting in a population of 986 neurons.

## Residual NMF method

As described in the main text, we first considered using a combination of non-negative least squares and non-negative matrix factorisation to decouple EA and SA (S1 Fig). We defined stimulus regressors $\boldsymbol{\phi}_i \in \mathbb{R}^T$ for $i = 1, \ldots, K$ by convolving the binary stimulus timeseries $\mathbf{s}_i \in \mathbb{R}^T$ with a calcium impulse response kernel $\mathbf{k}$ (a difference-of-exponentials function, defined below), giving $\boldsymbol{\phi}_i = \mathbf{k} * \mathbf{s}_i$. We then estimated regression coefficients $\beta_{ni}$ by solving the non-negative minimisation problem

$$\hat{\boldsymbol{\beta}}_n = \operatorname*{argmin}_{\boldsymbol{\beta}_n \geq 0} \left|\left| \mathbf{f}_n - \sum_{i=1}^{K} \beta_{ni} \boldsymbol{\phi}_i \right|\right|^2$$

where $\boldsymbol{\beta}_n = (\beta_{n1}, \ldots, \beta_{nK})^\top$. The evoked component of the fluorescence signal can be defined in terms of the estimated regression coefficients as

$$\hat{\mathbf{f}}^{\text{evoked}}_{\text{NNLS},n} = \hat{\boldsymbol{\beta}}_n^\top \boldsymbol{\Phi}$$

where $\boldsymbol{\Phi} = (\boldsymbol{\phi}_1^\top, \ldots, \boldsymbol{\phi}_K^\top)^\top$. Here NNLS refers to the non-negative least squares algorithm used to perform the minimisation. Residual data $\mathbf{e}_n$ was then defined as

$$\mathbf{e}_n = \sigma(\mathbf{f}_n - \hat{\boldsymbol{\beta}}_n^\top \boldsymbol{\Phi})$$

where $\sigma$ is a linear rectifier $\sigma(x) = \max(0, x)$ applied elementwise ensuring non-negativity of the residuals. We then applied NMF to the residual data $\mathbf{E} = (\mathbf{e}_1^\top, \ldots, \mathbf{e}_N^\top)^\top$ by solving the minimisation problem

$$\hat{\mathbf{W}}, \hat{\mathbf{H}} = \operatorname*{argmin}_{\mathbf{W}, \mathbf{H} \geq 0} \left|\left| \mathbf{E} - \mathbf{W}\mathbf{H} \right|\right|^2$$

where $\mathbf{W} \in \mathbb{R}^{N,L}$ and $\mathbf{H} \in \mathbb{R}^{L,T}$. The $L$ rows of $\hat{\mathbf{H}}$ are timeseries describing the evolution of low dimensional structure, and the columns of $\hat{\mathbf{W}}$ describe how neurons are coupled to such timeseries. The NMF-estimated SA for neuron $n$ is given by the projection of the latent timeseries onto a single dimension by the corresponding row of $\hat{\mathbf{W}}$

$$\hat{\mathbf{f}}^{\text{spont}}_{\text{NMF},n} = \hat{\mathbf{W}}_n \hat{\mathbf{H}}.$$

The full fluorescence data can thus be approximately reconstructed as

$$\mathbf{f}_n \approx \hat{\mathbf{f}}_{\text{NNLS},n}^{\text{evoked}} + \hat{\mathbf{f}}_{\text{NMF},n}^{\text{spont}}.$$

We use the NNLS routine in SciPy and the NMF routine in scikit-learn [44].

To evaluate the models of SA produced by NMF we then expressed the spontaneous components in terms of a basis of calcium impulses by solving the non-negative minimisation problem

$$\hat{\boldsymbol{\beta}}_n^{\text{ca}} = \underset{\boldsymbol{\beta}_n^{\text{ca}} \geq 0}{\text{argmin}} ||\hat{\mathbf{f}}_{\text{NMF},n}^{\text{spont}} - \sum_{t=0}^{T-1} \beta_{nt}^{\text{ca}} \boldsymbol{\psi}_t||^2$$

where each $\boldsymbol{\psi}_t$ is a calcium response following a unit impulse $\boldsymbol{\delta}_t^T$ at time $t$,

$$\boldsymbol{\psi}_t = \mathbf{k} * \boldsymbol{\delta}_t^T.$$

Here $\boldsymbol{\delta}_t^T$ is a vector of length $T$ that takes the value of 1 at $t$ and 0 elsewhere. The expression of $\hat{\mathbf{f}}_{\text{NMF},n}^{\text{spont}}$ in the basis of calcium transients is then given by $(\hat{\boldsymbol{\beta}}_n^{\text{ca}})^\top \boldsymbol{\Psi}$, where $\boldsymbol{\Psi} = (\boldsymbol{\psi}_0^\top, \ldots, \boldsymbol{\psi}_{T-1}^\top)^\top$. Note that this differs from the basis of stimulus regressors used to model the stimulus-driven component of neural activity as we employ a regressor for each time point $t$ in the entire trace.

## CILVA model

**Fluorescence model.** We model fluorescence data $f_n(t)$ as a linear transformation of the calcium concentration $c_n(t)$ plus independent and identically distributed additive Gaussian noise. The generative model for the observed fluorescence of neuron $n$ is thus

$$f_n(t) = \alpha_n c_n(t) + \beta_n + \epsilon_n(t), \tag{1}$$

$$\epsilon_n(t) \sim \mathcal{N}(0, \sigma_n^2) \tag{2}$$

where $\alpha_n$ is a scaling factor and $\beta_n$ is the baseline fluorescence level of neuron $n$. The assumption of Gaussian noise is a simple and tractable way to account for noise in both the calcium concentration and noise due to optical imaging. This model is standard for fluorescence imaging data [7, 23, 45].

**Calcium dynamics.** The calcium concentration $c_n(t)$ is generated as the convolution of a difference-of-exponentials kernel $\mathbf{k}$ with a function $\lambda_n$ that determines the intensity of neural activity,

$$c_n(t) = \sum_{\tau=0}^{t} k(t-\tau)\lambda_n(\tau). \tag{3}$$

The kernel $\mathbf{k}$ captures the stereotypical rise-and-decay calcium dynamics, which are assumed to possess time constants that are unchanging throughout the recording

$$k(t) = \exp(-t/\tau_d) - \exp(-t/\tau_r). \tag{4}$$

An explicit rise time was essential for modelling the experimental data with a GCaMP6s calcium indicator [6]. For the data used in the paper we used calcium transient time constants of $\tau_r = 5.68/F_s$ and $\tau_d = 11.5/F_s$, where $F_s = 2.1646$ Hz is the imaging rate of the fluorescence microscope for the zebrafish experiments and $F_s = 2.5$ Hz is the imaging rate for the mouse experiments. Below we also provide a penalised regression approach for estimating these time constants in necessary.

**Intensity function.**   Changes in the intracellular calcium concentrations are driven by an intensity function $\boldsymbol{\lambda}_n$ for each neuron $n$. We take advantage of the fact that we expect evoked responses to be time-locked to the presentation of a stimulus, with the remaining signal attributable to structured SA. The intensity is thus comprised of a stimulus drive and a latent drive

$$\lambda_n(t) = \mathbf{w}_n^\top \mathbf{s}(t) + \mathbf{b}_n^\top \mathbf{x}(t). \tag{5}$$

Here $\mathbf{w}_n \in \mathbb{R}^K$ corresponds to the stimulus filter for neuron $n$, $\mathbf{s}(t) \in \mathbb{R}^K$ is the vector describing which stimulus is active at time $t$ (under a 1-of-$K$ encoding scheme, such that $\mathbf{s}(t)$ has a 1 in index $i$ if the $i$th stimulus is active, and zeros elsewhere), $\mathbf{b}_n \in \mathbb{R}^L$ is a row of the factor loading matrix, and $\mathbf{x}(t) \in \mathbb{R}^L$ is the activity level of the latent factors at time $t$. Thus, at each time point the latent drive is the projection of a low dimensional latent process $\mathbf{x}(t)$ into one dimension. By fixing the onset of the stimulus drive and leaving the latent factors unconstrained, we allow the factors to adapt to the patterns of SA in the data.

Note that while evoked responses in our zebrafish data were well-described by a single impulse response, we may straightforwardly account for temporally extended responses by adjusting the stimulus design matrix **s** to include copies of each stimulus time shifted by single frames (up to a desired length) [24]. Moreover, trial-to-trial variability can arise in the evoked response itself through a multiplicative mechanism, rather than just additively via the spontaneous component. Incorporating a multiplicative component may slightly improve model fit but significantly complicates the inference for this model, however, and so we leave this for future work (also see Discussion).

**Latent factors.**   The latent factor activity $\mathbf{x}(t) \in \mathbb{R}^L$ lies in a lower dimensional subspace than the complete neural population activity $\mathbf{f}(t) \in \mathbb{R}^N$. Consequently, the variability that they account for in the model must be shared among groups of neurons. Without regularisation, the model faces identifiability issues because activity can be freely attributed to either the sensory stimuli or the latent factors. As the stimulus responses are already fixed to the observed stimulus times, we instead place a regularising exponential prior on the latent factors to encourage sparsity,

$$p(x_l(t)|\gamma) \propto \exp(-x_l(t)/\gamma) \tag{6}$$

for $0 \leq t \leq T - 1$ and $1 \leq l \leq L$. Here $\gamma$ is the parameterisation of the exponential distribution in terms of its mean; i.e., $\mathbb{E}[x_l(t)] = \gamma$, which acts as a sparsity penalty. Selection of $\gamma$ is described in the Model Selection section below. Since the latent variables are constrained to be non-negative, calculating the MAP estimate under the exponential prior is equivalent to maximising the log-likelihood with a lasso regularisier. Furthermore, while the $\Delta F/F$ that we are modelling can occasionally take negative values, this is considered to be a consequence of the imaging noise rather than a negative concentration of bound GCaMP. Thus our non-negativity constraint on the factor activity is in line with calcium imaging preprocessing methods that are based on non-negative deconvolution and non-negative matrix factorisation.

Note that we specifically do not enforce orthogonality constraints between the factor activity and stimulus times. As in Fig 3A, we wish to allow a combination of both latent factors and sensory stimuli to explain the observed fluorescence levels; the regularising prior acts to encourage the optimisation algorithm to explain calcium transients via sensory stimuli.

While many popular methods for analysing spike train data assume that latent factors obey a smooth temporal dynamics, our statistical model relies on the convolution of a sparse time-series of calcium influxes with a GCaMP kernel to generate the observed fluorescence signal. If instead factor activity states were constrained to vary smoothly and have high autocorrelation (e.g., under a Gaussian process prior), the predicted fluorescence transients would be

inaccurately prolonged following the GCaMP convolution. Indeed, calcium influx is typically well-described by sparse spike-and-slab models [46], and an exponential prior that specifically omits factor autocorrelation allows us to compromise between sparsity, model simplicity, and computational tractability.

**Evoked and spontaneous variance components.** Given the fitted model parameters, we defined the evoked and spontaneous components of the fluorescence signal as

$$\hat{\mathbf{f}}_n^{\text{evoked}} = \hat{\alpha}_n \mathbf{k} * \hat{\mathbf{w}}_n^\top \mathbf{s} + \hat{\beta}_n \mathbf{1}_T,$$

$$\hat{\mathbf{f}}_n^{\text{spont}} = \hat{\alpha}_n \mathbf{k} * \hat{\mathbf{b}}_n^\top \hat{\mathbf{x}} + \hat{\beta}_n \mathbf{1}_T.$$

Note that we include the baseline fluorescence term $\hat{\beta}_n$ to ensure the evoked and spontaneous traces are appropriately aligned with the raw fluorescence signal during visual comparisons. The variance of the reconstructed fluorescence levels can then be written in terms of these components

$$
\begin{aligned}
\text{var}[\hat{\mathbf{f}}_n] &= \text{var}[\hat{\alpha}_n \mathbf{k} * \hat{\mathbf{w}}_n^\top \mathbf{s} + \hat{\alpha}_n \mathbf{k} * \hat{\mathbf{b}}_n^\top \hat{\mathbf{x}}] \\
&= \text{var}[\hat{\mathbf{f}}_n^{\text{evoked}}] + \text{var}[\hat{\mathbf{f}}_n^{\text{spont}}] + 2\text{cov}[\hat{\mathbf{f}}_n^{\text{evoked}}, \hat{\mathbf{f}}_n^{\text{spont}}].
\end{aligned}
$$

When plotting variance components as proportions of sample variance as in Fig 4A, the sample variance $\text{var}[\mathbf{f}_n]$ is corrected for imaging noise by subtracting the estimated sample imaging noise variance $\sigma_n^2$. We then define the drive ratio for neuron $n$ using the variance components as

$$d_n = \frac{\text{var}[\hat{\mathbf{f}}_n^{\text{evoked}}] - \text{var}[\hat{\mathbf{f}}_n^{\text{spont}}]}{\text{var}[\hat{\mathbf{f}}_n^{\text{evoked}}] + \text{var}[\hat{\mathbf{f}}_n^{\text{spont}}]}.$$

This defines an index ranging from −1 to 1 that describes the extent to which a neuron is driven more by shared sources of SA or by EA.

Private variability can then be indirectly approximated by subtracting the estimated shared variance from the variance of the raw signal (corrected for imaging noise),

$$\text{priv}[\mathbf{f}_n] = \text{var}[\mathbf{f}_n] - \text{var}[\hat{\mathbf{f}}_n].$$

However, as $\text{var}[\hat{\mathbf{f}}_n]$ represents a lower bound on the shared variance, $\text{priv}[\mathbf{f}_n]$ only represents an upper bound on the private variance.

**Factor contribution index.** The contribution of a factor $\mathbf{x}_l$ is defined as the average reduction in explained correlation caused by removing factor $l$ from the model reconstruction of the fluorescence trace,

$$1 - \frac{1}{N}\sum_{n=1}^{N} \frac{\text{corr}[\mathbf{f}_n, \hat{\mathbf{f}}_{n(-l)}]}{\text{corr}[\mathbf{f}_n, \hat{\mathbf{f}}_n]}$$

where $\hat{\mathbf{f}}_{n(-l)} = \hat{\alpha}_n \mathbf{k} * (\hat{\mathbf{w}}_n^\top \mathbf{s} + \hat{\mathbf{b}}_{n(-l)}^\top \hat{\mathbf{x}}_{(-l)}) + \hat{\beta}_n \mathbf{1}_T$, and $\hat{\mathbf{b}}_{n(-l)}$ and $\hat{\mathbf{x}}_{(-l)}$ are obtained by deleting element $l$ and row $l$ from $\hat{\mathbf{b}}_n$ and $\hat{\mathbf{x}}$, respectively.

**Tuning curve comparison.** Tuning curves obtained by averaging were defined as the mean $\Delta F/F$ over the 4th to 7th frames following stimulus onset. As the stimulus filters $\{\hat{\mathbf{w}}_n\}$ are rescaled by our parameter identification algorithm (described below), we compared the averaging-based tuning curves with $k_{\max}\hat{\alpha}_n\hat{\mathbf{w}}_n$, where $k_{\max} = \max_t k(t)$ is the maximum value of the calcium kernel. This scaling of the stimulus filter reports the amplitudes of the calcium

transients evoked by each stimulus, which are directly comparable with the tuning curves obtained by averaging.

## Model fitting

We fit the model by maximising the posterior density of the latent variables,

$$\hat{\mathbf{x}}, \hat{\theta} = \underset{\mathbf{x}, \theta \geq 0}{\operatorname{argmax}} \ p(\mathbf{x}|\mathbf{f}, \theta, \gamma) = \underset{\mathbf{x}, \theta \geq 0}{\operatorname{argmax}} \ p(\mathbf{f}|\mathbf{x}, \theta)p(\mathbf{x}|\gamma) \tag{7}$$

where the parameters of the model are $\theta = (\{\alpha_n\}, \{\beta_n\}, \{\mathbf{w}_n\}, \{\mathbf{b}_n\})$. Ideally, one could perform this optimisation using the expectation-maximisation algorithm, which alternates between computing the posterior distribution over the latent factors $q(\mathbf{x}) = p(\mathbf{x}|\mathbf{f}, \theta, \gamma)$, and maximising the posterior expectation $\theta^{\text{new}} = \operatorname{argmax}_{\theta} \mathbb{E}_q[\ln \ p(\mathbf{f}, \mathbf{x}|\theta, \gamma)]$. However, the E-step is not analytically tractable since our exponential prior on $x_l(t)$ is non-conjugate for the likelihood model. Instead, we use a related "pseudo expectation-maximisation" approach [45] that alternately optimises Eq 7 according to the steps

$$\mathbf{x}^{(i+1)} = \underset{\mathbf{x} \geq 0}{\operatorname{argmax}} \ p(\mathbf{f}|\mathbf{x}, \theta^{(i)})p(\mathbf{x}|\gamma) \tag{8}$$

$$\theta^{(i+1)} = \underset{\theta \geq 0}{\operatorname{argmax}} \ p(\mathbf{f}|\mathbf{x}^{(i+1)}, \theta)p(\mathbf{x}^{(i+1)}|\gamma) \tag{9}$$

until numerical convergence or until $i$ reaches a user-specified number of iterations. The alternating maximisations are each performed using the bounded BFGS algorithm with limited memory (L-BFGS-B), with exact gradients derived below.

The logarithm of the joint model probability density is

$$\ln \ p(\mathbf{f}, \mathbf{x}|\theta, \gamma) = \sum_{t=0}^{T-1} \sum_{n=1}^{N} \ln \ p(f_n(t)|\mathbf{x}(0), \dots, \mathbf{x}(t), \theta) + \sum_{t=0}^{T-1} \sum_{l=1}^{L} \ln \ p(x_l(t)|\gamma) + \text{constant}$$

where the constant term does not depend on the parameters of $\theta$ to be estimated. Let $\ell(\mathbf{x}, \theta) = \ln p(\mathbf{f}, \mathbf{x}|\theta, \gamma)$, and let $\mathcal{E}_n(t)$ denote the model reconstruction error for neuron $n$ in imaging frame $t$,

$$\mathcal{E}_n(t) = f_n(t) - \alpha_n(\mathbf{k} * \boldsymbol{\lambda}_n)(t) - \beta_n.$$

The derivatives of $\ell(\mathbf{x}, \theta)$ with respect to the parameters and latent variables are then

$$\frac{\partial}{\partial \alpha_n} \ell(\mathbf{x}, \theta) = \frac{\alpha_n}{\sigma_n^2} \sum_{t=0}^{T-1} \mathcal{E}_n(t) \cdot (\mathbf{k} * \boldsymbol{\lambda}_n)(t)$$

$$\frac{\partial}{\partial \beta_n} \ell(\mathbf{x}, \theta) = \frac{1}{\sigma_n^2} \sum_{t=0}^{T-1} \mathcal{E}_n(t)$$

$$\frac{\partial}{\partial \mathbf{w}_n} \ell(\mathbf{x}, \theta) = \frac{\alpha_n}{\sigma_n^2} \sum_{t=0}^{T-1} \mathcal{E}_n(t) \cdot (\mathbf{k} * \mathbf{s})(t)$$

$$\frac{\partial}{\partial \mathbf{b}_n} \ell(\mathbf{x}, \theta) = \frac{\alpha_n}{\sigma_n^2} \sum_{t=0}^{T-1} \mathcal{E}_n(t) \cdot (\mathbf{k} * \mathbf{x})(t)$$

$$\frac{\partial}{\partial \mathbf{x}(\tau)} \ell(\mathbf{x}, \theta) = \sum_{n=1}^{N} \sum_{t=\tau}^{T-1} \frac{\alpha_n}{\sigma_n^2} \mathbf{b}_n \mathcal{E}_n(t) k(t - \tau) - \frac{1}{\gamma} \mathbf{1}_L$$

where for a matrix $\mathbf{\Lambda} \in \mathbb{R}^{T \times q}$ the convolution $\mathbf{k} * \mathbf{\Lambda} \in \mathbb{R}^{T \times q}$ is performed row-wise. In practice we vectorise the computation of the gradients to improve efficiency.

The imaging noise variance terms $\sigma_n^2$ are estimated using the method in ref. [23]. Specifically, $\sigma_n^2$ is estimated as the mean of the power spectral density of $\mathbf{f}_n$ over the range $(F_s/4, F_s/2)$, where $F_s$ is the imaging rate of the fluorescence microscope.

**Model identifiability.** As is common in factor analysis-style methods, the model parameters and latent variables $(\theta, \mathbf{x})$ are not uniquely identifiable in their current form. Our model-fitting algorithm thus transforms the estimates $(\hat{\theta}, \hat{\mathbf{x}})$ into a standardised form according to the following procedure. First we fit the CILVA model to data $\{\mathbf{f}_n\}$ using the MAP estimator to obtain model parameters $\{\hat{\alpha}_n\}, \{\hat{\beta}_n\}, \{\hat{\mathbf{w}}_n\}, \{\hat{\mathbf{b}}_n\}, \{\hat{\mathbf{x}}_l\}, \{\hat{\sigma}_n^2\}$. We then sort factors $\hat{\mathbf{x}}_l$ in descending order of their Euclidean norm so that $||\hat{\mathbf{x}}_1|| \geq \cdots \geq ||\hat{\mathbf{x}}_L||$, and sort factor coupling column vectors $\hat{\mathbf{b}}^{(l)} \in \mathbb{R}^N$ to take the same order. Next, we normalise latent factors and proportionally rescale factor coupling vectors,

$$(\hat{\mathbf{x}}_l, \hat{\mathbf{b}}^{(l)}) \leftarrow (\frac{1}{||\hat{\mathbf{x}}_l||}\hat{\mathbf{x}}_l, ||\hat{\mathbf{x}}_l||\hat{\mathbf{b}}^{(l)}).$$

The latent factors are now identifiable. Finally, we normalise the static model parameters by the norm of the neural intensity vector,

$$(\hat{\alpha}_n, \hat{\mathbf{w}}_n, \hat{\mathbf{b}}_n) \leftarrow (||\hat{\boldsymbol{\lambda}}_n||\hat{\alpha}_n, \frac{1}{||\hat{\boldsymbol{\lambda}}_n||}\hat{\mathbf{w}}_n, \frac{1}{||\hat{\boldsymbol{\lambda}}_n||}\hat{\mathbf{b}}_n).$$

This ensures identifiability of the static model parameters $\theta$.

**Parameter initialisation.** We also implemented a simple penalised regression approach to estimate the calcium transient time constants $\tau_r$ and $\tau_d$ if required. The idea is to alternately estimate tuning curves (using knowledge of the stimulus presentation times) and update our time constants given these new tuning curves. The constants $\tau_r$ and $\tau_d$ must respect the inequality

$$0 < \tau_r < \tau_d.$$

We thus parameterise $\tau_d$ in terms of the rise time constant and a positive offset,

$$\tau_d = \tau_r + \Delta, \ \ \text{where } \Delta > 0.$$

Let $k_{\tau_r, \Delta}(t) = \exp(-t/(\tau_r + \Delta)) - \exp(-t/\tau_r)$. Given some values of $\tau_r$ and $\Delta$, we define $\mathbf{\Phi}_{\tau_r, \Delta} \in \mathbb{R}^{K \times T}$ analogous to the residual NMF method with

$$(\mathbf{\Phi}_{\tau_r, \Delta})_i = \mathbf{k}_{\tau_r, \Delta} * \mathbf{s}_i$$

for $i = 1, \ldots, K$. For every neuron $n$ we then fit tuning curves as

$$\hat{\boldsymbol{\omega}}_n = \underset{\tau_r, \Delta > 0}{\operatorname{argmin}} ||\mathbf{f}_n - \boldsymbol{\omega}_n^\top \mathbf{\Phi}_{\tau_r, \Delta}||^2$$

using non-negative least squares. Then, given a set of tuning curves $\{\hat{\boldsymbol{\omega}}_n\}$, we update the time constants by minimising the model reconstruction error averaging over all neurons,

$$\hat{\tau}_r, \hat{\Delta} = \underset{\tau_r, \Delta > 0}{\operatorname{argmin}} \left\{ \frac{1}{2} \sum_{n=1}^{N} \sum_{t=0}^{T-1} (f_n(t) - (\mathbf{k}_{\tau_r, \Delta} * \hat{\boldsymbol{\omega}}_n^\top \mathbf{s})(t))^2 + \eta(\tau_r + \Delta) \right\}$$

where $\eta > 0$ is a chosen penalty coefficient. The derivative of this term with respect to $\tilde{\tau} \in \{\tau_r, \Delta\}$ is

$$-\sum_{n=1}^{N}\sum_{t=0}^{T-1}(f_n(t) - (\mathbf{k}_{\tau_r,\Delta} * \hat{\boldsymbol{\omega}}_n^\top \mathbf{s})(t)) \cdot \left(\frac{\partial}{\partial \tilde{\tau}}\mathbf{k}_{\tau_r,\Delta} * \hat{\boldsymbol{\omega}}_n^\top \mathbf{s}\right) + \eta$$

where

$$\frac{\partial}{\partial \tau_r}k_{\tau_r,\Delta}(t) = \frac{t}{(\tau_r + \Delta)^2}\exp\left(\frac{-t}{\tau_r + \Delta}\right) + \frac{t}{\tau_r^2}\exp\left(\frac{-t}{\tau_r}\right),$$

and

$$\frac{\partial}{\partial \Delta}k_{\tau_r,\Delta}(t) = \frac{t}{(\tau_r + \Delta)^2}\exp\left(\frac{-t}{\tau_r + \Delta}\right).$$

We perform the non-negative minimisation with these gradients using L-BFGS-B. Learning the time constants typically only required several alternations of estimating the tuning curves $\{\hat{\boldsymbol{\omega}}_n\}$ and updating the time constants $(\hat{\tau}_r, \hat{\Delta})$.

We use the stimulus regressors to also initialise the filters $\mathbf{w}_n$; i.e.,

$$\mathbf{w}_n^{\text{init}} = \underset{\mathbf{w}_n \geq 0}{\text{argmin}} ||\mathbf{f}_n - \mathbf{w}_n^\top \boldsymbol{\Phi}||^2$$

with the calcium time constants obtained either by the penalised regression approach described above or by manual specification. We then initialise $\alpha_n$ as a small perturbation around 1, $\alpha_n^{\text{init}} \sim \mathcal{N}(1, 10^{-2})$, and $\beta_n^{\text{init}} = 0$. We initialise the latent factor coupling strengths as uniform samples from the unit interval, $b_{nl}^{\text{init}} \sim U(0, 1)$, and the factor activity levels uniformly from a small interval, $x_l^{\text{init}}(t) \sim U(0, 1/5)$.

**Model selection.** CILVA depends on two key hyperparameters: the number of latent factors $L$ and the sparsity parameter $\gamma$. Here we describe how we estimate these hyperparameters. To avoid local minima, we fit the model several times to the data with different random initialisations of the factor coupling vectors $\{\mathbf{b}_n\}$ and factor activities $\{\mathbf{x}_l\}$. For a given $L$ and $\gamma$ we fit the latent variables and parameters on training data as

$$\hat{\mathbf{x}}_{L,\gamma}^{(i)}, \hat{\theta}_{L,\gamma}^{(i)} = \underset{(\mathbf{x}_{L,\gamma}^{(i)}, \theta_{L,\gamma}^{(i)}) \geq 0}{\text{argmax}} p(\mathbf{f}|\mathbf{x}_{L,\gamma}^{(i)}, \theta_{L,\gamma}^{(i)})p(\mathbf{x}_{L,\gamma}^{(i)}|\gamma)$$

where $i = 1, \ldots, i_{\max}$ denotes the $i$th initialisation of $\mathbf{x}$ and $\theta$. We select the optimal parameters $\theta_{L,\gamma}^{(i)}$ and hyperparameter $\gamma$ as those that maximise the joint density of the data and latent variables on 5 minutes of held-out test data $\mathbf{f}^{\text{test}}$; i.e.,

$$\hat{\theta}_L, \hat{\gamma}_L = \underset{(\theta_{L,\gamma}^{(i)}, \gamma)}{\text{argmax}} p(\mathbf{f}^{\text{test}}|\hat{\mathbf{x}}_{L,\gamma}^{\text{test}}, \hat{\theta}_{L,\gamma}^{(i)})p(\hat{\mathbf{x}}_{L,\gamma}^{\text{test}}|\gamma),$$

where values of $\gamma$ are obtained via grid search over a small interval $[\Delta_\gamma, d\Delta_\gamma]$ with step-size $\Delta_\gamma$ and number of grid points $d$. Here we infer new latent variables $\hat{\mathbf{x}}_{L,\gamma}^{\text{test}}$ that explain the patterns of spontaneous activity in the test data $\mathbf{f}^{\text{test}}$. The selected latent variables for the training data are then those that correspond to the optimal $\theta$ and $\gamma$,

$$\hat{\mathbf{x}}_L = \underset{\mathbf{x} \geq 0}{\text{argmax}} p(\mathbf{f}|\mathbf{x}, \hat{\theta}_L)p(\mathbf{x}|\hat{\gamma}_L).$$

We used $\Delta_\gamma = 0.2$, $d = 10$ and $i_{\max} = 5$. For the example zebrafish used in the main text we

selected $L$ as the number of factors after which the mean correlation coefficient between the raw fluorescence traces and model-reconstructions failed to substantially improve (i.e., at the 'elbow' in Fig 3D).

Fitting CILVA for testing and preliminary data analysis required a computation time of $\sim 10$ minutes on a 64-bit MacBook Pro with a 3.1 GHz Intel Core i7 Processor and 8 GB DDR3 RAM running Python 3.6.4. For the model fits in this paper we allowed the optimisation procedure to run to a user-specified number of alternations of Eqs 8 and 9 (typically $\sim 100$), performed on a computer cluster with 17 Dell EMC PowerEdgeR740 compute nodes, each comprised of two Intel Xeon Gold 6132 processors with 384 GB DDR4 RAM. Scheduled jobs were allocated 2 CPUs and 5GB RAM, and required $\sim 1$ hour to complete.

## Simulated data

To generate simulated data we sampled the latent factors from a zero-inflated exponential distribution with probability $\xi$ of a non-zero latent event,

$$x_l(t) \sim (1 - \xi)\delta(x_l(t)) + \xi \mathrm{Exp}(\gamma_x).$$

This ensured the latent factor activity was sparse. We also introduced a private SA term $z_n(t)$ for neuron $n$ at time $t$ by sampling from a zero-inflated exponential with probability $\pi$ of a non-zero private event,

$$z_n(t) \sim (1 - \pi)\delta(z_n(t)) + \pi \mathrm{Exp}(\gamma_z).$$

The intensity function was then given by

$$\lambda_n(t) = \mathbf{w}_n^\top \mathbf{s}(t) + \mathbf{b}_n^\top \mathbf{x}(t) + z_n(t)$$

with the fluorescence levels following the standard CILVA model with a common imaging noise variance $\sigma^2$,

$$f_n(t) = \alpha_n(\mathbf{k} * \lambda_n)(t) + \beta_n + \epsilon_n(t)$$
$$\epsilon_n(t) \sim \mathcal{N}(0, \sigma^2).$$

We sampled $\alpha_n$ from the discrete uniform distribution on $\{2, \ldots, 10\}$ and for simplicity set $\beta_n = 0$. The tuning curves $\mathbf{w}_n$ were defined as Gaussian functions $x \mapsto \exp(-(x - \mu_n)^2/2\nu)$. For the simulation of data in response to well-spaced, low dimensional stimuli (cf. Fig 3) we sampled the centres $\mu_n$ uniformly from the interval $[0, K]$, where $K$ is the number of stimuli, and sampled the widths $\nu_n$ uniformly from $[0, K/2]$. For the simulation of data in response to rapidly presented, high dimensional stimuli (cf. Fig 5) we chose our receptive fields to be more selective and sampled $\nu_n$ uniformly from $[0, \sqrt{K}]$.

Factor coupling vectors $\mathbf{b}_n$ were defined by evenly assigning the $N$ neurons to $L$ factors, and sampling $b_{nl} \sim U[q, 1]$ if neuron $n$ is assigned to factor $l$, and $b_{nl} \sim U[0, 1 - q]$ otherwise. We found $q = 0.85$ provided simulations that appeared similar to the experimental data. We characterised the model reconstruction quality in terms of $\pi$ and $\sigma^2$ in S2 and S3 Figs, with the associated model parameters provided in S1 and S2 Tables.

## Supporting information

**S1 Video. Reconstructed calcium imaging data from the larval zebrafish optic tectum with stimulus and inferred factor activity.** 143 neurons recorded from the optic tectum in response to 9 visual stimuli. Latent factors explain the presence of structured patterns of spontaneous activity between stimulus onset times. Stimulus and factor activity have been

convolved with a GCaMP6s calcium kernel for improved visual comparison between stimuli, factors, and neural activity. Individual neuron intensities are normalised to range from 0 to 1. This data corresponds to 5 minutes of activity from the example zebrafish in Figs 1–4.
(MP4)

**S2 Video. Decoupled evoked and spontaneous activity from the larval zebrafish optic tectum.** Decomposition of the activity in S1 Video into its evoked and spontaneous components.
(MP4)

**S1 Table. Parameters for simulated data corresponding to the presentation of a low dimensional stimulus with prolonged interstimulus intervals (analogous to the zebrafish data).** Parameters related to time are defined with respect to imaging rate. Listed values of $\pi$ and $\sigma^2$ are defaults, but are varied over the range specified in parentheses.
(PDF)

**S2 Table. Parameters for simulated data with rapid presentation of a high dimensional stimulus (analogous to the mouse data).** Parameters related to time are defined with respect to imaging rate. Listed values of $\pi$ and $\sigma^2$ are defaults, but are varied over the range specified in parentheses.
(PDF)

**S3 Table. Numerical values for the histograms in S11 Fig.** Zebrafish 5 corresponds to the example used in Figs 1–4 and S1 and S4–S7 Figs.
(PDF)

**S1 Fig. Residual NMF approach to decoupling EA and SA in larval zebrafish optic tectum.** (A) Fluorescence traces from 10 example neurons. Dashed vertical lines indicate stimulus onset; colour represents azimuth angle of presented stimulus. (B) Example fluorescence trace segment illustrating that spontaneous calcium transients can occur just before stimulus onset. (C) A simple estimate of the stimulus-driven component of population data can be obtained by multiple regression of fluorescence traces onto stimulus regressors using non-negative least squares. (D) After estimating the stimulus-driven component, low dimensional structure in the residual data can be estimated using non-negative matrix factorisation. (E) Patterns of SA shared between groups of neurons found via NMF. For consistency with later results, we here applied NMF with three latent factors. Each row corresponds to the activity of one factor. (F) Top: component of the raw fluorescence trace (black) considered to be SA by the residual NMF approach (blue). NMF often produces estimates with erratic and sudden changes in calcium levels that fail to respect the stereotypical structure of calcium activity. Bottom: additional examples of shared SA estimated from the residuals using NMF (blue). For comparison, the same estimates are shown when expressed in a basis of calcium impulse response functions located at each time point (orange, Methods). Deviations from the orange curve demonstrate atypical calcium behaviour. Samples were selected for illustration from among the 10 neurons best explained by the residual NMF approach.
(PDF)

**S2 Fig. Results on simulated data (analogous to the zebrafish data).** To validate performance we fit the model to simulated data (see Methods). The two primary constraints on model performance are (i) the rate $\pi$ of private spontaneous events, and (ii) the variance $\sigma^2$ of the imaging noise. We systematically varied these two parameters and observed the ability of the model to recover the underlying evoked and spontaneous components. Parameters used in the simulations are given in S1 Table. (A) Ten randomly chosen neurons from an example simulation with $\pi = 0.05$ and $\sigma^2 = 0.1$. Black traces show simulated raw fluorescence data. The

true composition of the fluorescence trace is given in red (EA) and blue (shared SA). (B) The correlation coefficient between the raw fluorescence trace and model reconstruction decreases as the rate of private spontaneous events increases. (C) Histograms of correlation coefficients for three example values of $\pi$. (D), While the correlation coefficient decreases with $\pi$, recovery of the evoked (left) and spontaneous (right) fluorescence components remains highly accurate. (E)—(G) Same as (B)—(D) but with varying noise variances $\sigma^2$. High noise variances limit the correlation between the raw (noisy) fluorescence trace and (noiseless) model reconstruction, but recovery of the evoked and spontaneous components is still very robust. All shaded regions represent 95th percentiles.
(PDF)

**S3 Fig. Results on simulated data with rapidly presented high-dimensional stimuli (analogous to the mouse data).** Parameters used in the simulations are given in S2 Table. (A) Ten randomly chosen neurons from an example simulation with $\pi = 0.05$ and $\sigma^2 = 0.1$. Black traces show simulated raw fluorescence data. The true composition of the fluorescence trace is given in red (EA) and blue (shared SA). (B) The correlation coefficient between the raw fluorescence trace and model reconstruction decreases as the rate of private spontaneous events increases. (C) Histograms of correlation coefficients for three example values of $\pi$. (D) While the correlation coefficient decreases with $\pi$, recovery of the evoked (left) and spontaneous (right) fluorescence components remains highly accurate. (E)—(G) Same as (B)—(D) but with varying noise variances $\sigma^2$. All shaded regions represent 95th percentiles.
(PDF)

**S4 Fig. Consistency of modelling outcomes with CaImAn preprocessing of zebrafish in Figs 3 and 4.** (A) Results of fitting CILVA and decoupling EA (red) and shared SA (blue) in an experimental recording with CaImAn preprocessing (cf. Fig 3A). Inset numbers denote the Pearson correlation coefficient between raw fluorescence trace and model fit. The 10 neurons with the highest correlations between data and model fit are shown. (B) Distribution of correlation coefficients between data and model fits (cf. Fig 3B). Shuffled data obtained by cyclically permuting each trace by a random offset while preserving its temporal structure. (C) Estimated factor coupling matrix shows that latent factors target distinct, non-overlapping sets of neurons. (D) Spatial organisation of latent factors underlying SA (cf. Fig 4G). The three non-overlapping factors are spatially localised and tile the imaging plane. (E) Spatial organisation of the evoked and spontaneous variance components (cf. Fig 4H and 4I). Cell opacity is proportional to the fraction of variance attributable to EA or SA for the given neuron.
(PDF)

**S5 Fig. Model fits for 35 neurons sampled from the larval zebrafish in Fig 3.** Example fluorescence traces (black) and corresponding model fits (green). Dashed vertical lines indicate stimulus onset times. Inset numbers denote Pearson correlation coefficient between raw trace and model fit. Sampled neurons are sorted by correlation. Poor fits can result from neurons that show inconsistent responses (or no responses) to presented stimuli or neurons dominated by private SA (and therefore that cannot be assigned to a latent factor). Another potential reason the model would fit poorly is segmentation errors when identifying neurons. However, manual inspection of the raw data suggested that this was not the case for the neurons shown here.
(PDF)

**S6 Fig. Decoupling of evoked (red) and spontaneous (blue) calcium transients corresponding to the neurons from S5 Fig.**
(PDF)

**S7 Fig. Residuals corresponding to the neurons from S5 Fig.** Residual data obtained by subtracting model fit from the raw data (i.e. $\mathbf{f}_n - \hat{\mathbf{f}}_n$). Inset numbers denote the correlation coefficients from the model fits in S5 Fig. Ideal residuals appear as independent and identically distributed samples from a Gaussian noise distribution. Systematic deviations from Gaussian noise reflect calcium transients not captured by the model, and contribute to measurements of private variability.
(PDF)

**S8 Fig. Decoupling of evoked and spontaneous activity corresponding to neurons in Fig 3H.** Neurons ordered the same as Fig 3H, with the neuron marked by an asterisk (11th trace) corresponding to the similarly marked neuron in Fig 3H.
(PDF)

**S9 Fig. Model fit from a second zebrafish demonstrating similar features to fish shown in the main text.** (A) Example fluorescence traces (black) and model fits (green) for the twelve best fitting neurons. Inset numbers denote the Pearson correlation coefficient between raw trace and model fit. (B) Application of the statistical model to decouple EA (red) and shared SA (blue). (C) Distribution of correlation coefficients between data and model fits. Shuffled data (gray) obtained by cyclically permuting each model fit by a random offset while preserving its temporal structure. (D) Inferred latent factor timeseries. Inset numbers denote the factor contribution indices. (E) Factor coupling matrix. (F) Cumulative factor contribution indices for 0-3 latent factors. (G) Correlation coefficient between raw fluorescence trace and model fit with and without incorporation of SA. Neurons with strongly negative drive ratios show marked improvement in quality of model fit. (H) Cross-correlograms show little interaction between latent factors. (I) Example stimulus filters (red). Tuning curves obtained by averaging fluorescence levels over a small window following stimulus presentation provided for comparison (gray). Shaded error bars represent one standard deviation. (J) Retinotopic maps obtained by averaging (left) and by fitting CILVA (right).
(PDF)

**S10 Fig. Model fit from a third zebrafish.** (A) Example fluorescence traces (black) and model fits (green) for the twelve best fitting neurons. Inset numbers denote the Pearson correlation coefficient between raw trace and model fit. (B) Application of the statistical model to decouple EA (red) and shared SA (blue). (C) Distribution of correlation coefficients between data and model fits. Shuffled data (gray) obtained by cyclically permuting each model fit by a random offset while preserving its temporal structure. (D) Inferred latent factor timeseries. Inset numbers denote the factor contribution indices. (E) Factor coupling matrix. (F) Cumulative factor contribution indices for 0-3 latent factors. (G) Correlation coefficient between raw fluorescence trace and model fit with and without incorporation of SA. Neurons with strongly negative drive ratios show marked improvement in quality of model fit. (H) Cross-correlograms show little interaction between latent factors. (I) Example stimulus filters (red). Tuning curves obtained by averaging fluorescence levels over a small window following stimulus presentation provided for comparison (gray). Shaded error bars represent one standard deviation. (J) Retinotopic maps obtained by averaging (left) and by fitting CILVA (right).
(PDF)

**S11 Fig. Consistency of CILVA fits across a population of zebrafish larvae.** Black triangles point to the example fish from the main text. (A) Mean and interquartile range (IQR) of correlation coefficient distributions for $n = 8$ larvae. (B) Distribution of factor contribution indices. For model fits with 3 latent sources of SA, each factor has a contribution index of $\sim 0.1$. (C)

Distribution of fraction of neurons 'shared' between multiple factors. Neurons were considered shared if they were coupled to more than one factor with coupling strengths exceeding a threshold of 25% of the maximum coupling strength for that factor. (D) Mean improvement in correlation coefficients with incorporation of latent sources of SA. (E) Distribution of mean drive ratios across the population of larvae, centered at −0.01, suggesting that SA and EA are largely balanced within individual fish. (F) The mean absolute values of the drive ratio are greater than 0, showing that individual neurons tend to be biased towards either EA or SA. Histograms in panels B—F obtained by non-parametric density estimation with Gaussian kernels. Raw data points used for histograms given in S3 Table.
(PDF)

**S12 Fig. Application of CILVA to mouse visual cortex.** (A) We fit the model with 10 latent factors. While the contribution indices for the factors gradually diminished (panel D), varying the number of factors from 1 to 20 did not identify a point at which the overall quality of fit failed to increase, including in held-out test data. (B) Cross-correlograms between latent factor timeseries indicate factors underlying SA are mutually independent. (C) Cumulative contribution of factors to quality of model fit. (D) Correlation coefficients between raw fluorescence trace and model fit with and without the SA component. Neurons with negative drive ratios (blue circles) demonstrate substantial improvement in the quality of model fit when incorporating SA. (E) Improvement in the quality of model fit when incorporating the SA component is statistically significant ($p < 0.001$, Wilcoxon signed-rank test). (F) Example decoupling of EA and SA for the 30 best fit neurons (top) and underlying latent factor timeseries (bottom). (G) Close-up of model fit from neurons in dashed region in panel G. Inset numbers denote Pearson correlation between raw data and full model fit.
(PDF)

## Acknowledgments

We thank Carsen Stringer for helpful feedback on an earlier version of the paper, and Robert Wong for assistance with data preprocessing.

## Author Contributions

**Conceptualization:** Marcus A. Triplett, Geoffrey J. Goodhill.

**Data curation:** Zac Pujic, Biao Sun, Lilach Avitan.

**Formal analysis:** Marcus A. Triplett.

**Funding acquisition:** Geoffrey J. Goodhill.

**Investigation:** Marcus A. Triplett, Geoffrey J. Goodhill.

**Methodology:** Marcus A. Triplett.

**Project administration:** Geoffrey J. Goodhill.

**Resources:** Zac Pujic, Biao Sun, Lilach Avitan.

**Software:** Marcus A. Triplett.

**Supervision:** Geoffrey J. Goodhill.

**Validation:** Marcus A. Triplett.

**Visualization:** Marcus A. Triplett.

**Writing – original draft:** Marcus A. Triplett, Geoffrey J. Goodhill.

**Writing – review & editing:** Marcus A. Triplett, Zac Pujic, Biao Sun, Lilach Avitan, Geoffrey J. Goodhill.

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
