## [Decision Letter · Decision Letter 0]

12 Aug 2020

Dear Dr. Goodhill,

Thank you very much for submitting your manuscript "Model-based decoupling of evoked and spontaneous neural activity in calcium imaging data" for consideration at PLOS Computational Biology. As with all papers reviewed by the journal, your manuscript was reviewed by members of the editorial board and by several independent reviewers. The reviewers appreciated the attention to an important topic. Based on the reviews, we are likely to accept this manuscript for publication, providing that you modify the manuscript according to the review recommendations.

Sincerely,

Saad Jbabdi

Associate Editor

PLOS Computational Biology

Kim Blackwell

Deputy Editor

PLOS Computational Biology

[LINK]

Reviewer's Responses to Questions

**Comments to the Authors:**

Reviewer #1: The authors present a useful and timely method for distinguishing spontaneous and evoked activity in calcium imaging data. This enables more accurately assessing spontaneous and evoked activity, which will always occur simultaneously in any in vivo recording setting. The authors should be congratulated in particular for the clarity of presentation and writing. I have some minor questions and points that would be helpful to clear up prior to publication but generally find the publication is acceptable for publication.

MINOR ISSUES / QUESTIONS

What causes neurons to have a poor model fit, eg the neurons at the left side of the red distribution in Fig 3B? I am curious if the authors have inspected these neurons to see if they might be poorly sampled in the imaging data, improperly segmented, and/or otherwise ‘bad’ in a way that might explain the poor performance of the model. My goal by asking this is to improve the work yet further by attempting to understand why it does not work when the model fit is poor. The authors mention in a few places that the neurons that are not fit well may be generally quiescent, unresponsive to sensory stimuli, and/or driven by some spontaneous activity not shared with other neurons recorded. But perhaps the issue is not the model but the data?

Is it possible to show the raw traces for a neuron with an inferred tuning that is very different from the one obtained by averaging as showing in H? The raw traces in 3A are very useful for understanding how the CILVA approach helps. And the tuning curves shown in H show some substantial differences. Is it possible some of the neurons in 3A happen to be some of the neurons in 3H already? I am curious for example about the neuron in the second from the bottom row and second from the last column in 3H — this is one place where the approach seems to massively beat the standard approach!

Line 159-161: “we encouraged sparsity by placing a non-negative prior on the latent factors with high density near zero, and used a simple model selection procedure to estimate the sparsity penalty” I’m curious why the authors did not use lasso or ridge to enforce sparsity. This also effects the model fitting possibilities (from Line 533).

MINOR TYPOGRAPHICAL/CLARITY ISSUES

Fig 1B bottom left stimulus colour code seems redundant as it provides no extra information given the order of stimuli can be inferred from the sequence of coloured dashed vertical lines.

Lambda is referred to as “rate of calcium influx” in line 143, “underlying intensity of neural activity” in Fig 2A legend, and finally “Intensity functions” in Fig 2B legend. It is bold in Fig 2 legend but not Fig 2 itself or line 143. Please describe and use consistently.

Line 238 “well-describes” odd phrasing

“Table S3: Raw data points for the histograms in Figure 5.” Does not appear to refer to Figure 5

It seems odds to present the grand average data in S10 rather than a main figure? My opinion is not strong but if other reviewers make a similar remark perhaps moving it to a main figure would be appropriate.

Reviewer #2: The authors developed a method to fit simultaneously the evoked and “spontaneous” components of temporal fluctuations in population activity. As the authors acknowledge, the particular model used is related to previous “factor analysis-type” models, and/but includes a positivity constraint on the factors and no temporal structure. Using this model, the authors describe the dynamics of population activity in the optic tectum of zebrafish and in mouse visual cortex.

The paper is clear, well written and very polished. Steps in the algorithm are clearly explained and so are the results of the analyses. The fact that the code is available and that the method is relatively straightforward might actually motivate researchers to try using the model, which would be a big plus for the authors. I don’t have any major issues with the manuscript.

Comments.

1) Dynamics of the latent factors. In the traces in Fig3A it looks like several spontaneous “bumps” are evoked by the stimulus. As far as I can tell, this tendency is not quantified except for the size of the orange ‘covariance’ bars in Fig4A. Somehow looking at 4A it seems like the covariance is marginal, but then looking at 3A it’s easy to see ‘by eye’ the evoked factor transients. It should be quite straightforward for the authors to quantify these evoked factor transients.

More generally, what are the implications for the model of a situation where spontaneous events are ‘triggered’ by the stimulus? In the cortex, phenomena like this have been well characterised in the dynamics of e.g., up-states (Luczak & Harris, Hasenstaub & McCormick, etc). Up-states can and do occur spontaneously, but they are easily evoked. When the interval of stimulus presentation is regular (as in the current study?), the timing of spontaneous population events has even been shown to track (not presented) stimuli! (Li, Jingcheng, et al. "Primary auditory cortex is required for anticipatory motor response." Cerebral Cortex 27.6 (2017): 3254-3271).

Conceivably, every (or a large majority of) spike comes from these population events and the stimulus simply triggers them sometimes. Is this scenario describable by the model? Somehow the additive nature of the interaction between stimulus and factors does not lend itself easily to describe this scenario in my mind, but maybe I’m missing something? Could the authors elaborate on this point?

2) It would be useful to mention explicitly the part of the variability which is private when describing the model. Although private variability is mentioned several times (pages 12, 20,21…) I could not find a mention to it in the description of the formulas (only for simulated data)

3) Positivity constraints. Given that the authors are modelling DeltaF/F, which can be negative, perhaps a comment can e made on the virtues of the positivity constraint on the factors?

4) Autocorrelation of the factors. I could not understand the argument given for justifying the lack of autocorrelation of the factors. While certainly simplicity is an argument (and a good one if the model generally works well), the authors rather point to a possible ambiguity between the time-scale of the autocorrelation of the factors and that of the fluorescence due to the Ca-dynamics. I must have missed something, because the Ca-dynamics is already explicitly built into the model through the kernel k. Looking at the autocorrelations of the factors in, e.g, Fig3G, it seems like sometimes there are slow dynamics (for this fish in factor 3). This is not to say that assuming no autocorrelation is a negative feature per se. I just didn’t understand if the motivation was simplicity or something else.

5) Fig3G and equivalent panels in other figures. The y-lim in these plots (set to include the peak at zero) is unfortunate, as it prevents the reader from assessing the temporal structure of the signals. In some cases like factor 2 if FigS9H and others, there even seems to be some oscillatory structure (what is the period of this oscillation? How does it compare to the inter-stimulus interval?)

6) Topography of the dynamics. The results in Fig4G-I are nice, but I didn’t understand the sorting of neurons in 4A. The legend says neurons are sorted by A-P coordinate, but 4A is not nearly as ordered and structured as 4G. Given that the shape of the analysed activity is effectively 1D (although slightly curved), can neurons in 4A be sorted according to this 1D axis? If so, we would see the factor loadings in 4A bottom nicely ordered.

7) Fig4C. Again, this format seems not ideal for the info that this plot is supposed to convey (maybe a scatter plot like 4C?). Looking at 4C, the difference in covariance looks marginal. It seems to me like it should be possible to test the hypothesis that the covariance in the sample is significantly larger than that of a null model for each neutron.

8) Fig 4F. I think a violin plot would make this figure nicer and more transparent.

**Have all data underlying the figures and results presented in the manuscript been provided?**

Reviewer #1: Yes

Reviewer #2: None

PLOS authors have the option to publish the peer review history of their article (what does this mean?). If published, this will include your full peer review and any attached files.

Reviewer #1: **Yes: **Adam M. Packer

Reviewer #2: No
---

## [Editor Report · Decision Letter 1]

10 Sep 2020

Dear Dr. Goodhill,

We are pleased to inform you that your manuscript 'Model-based decoupling of evoked and spontaneous neural activity in calcium imaging data' has been provisionally accepted for publication in PLOS Computational Biology.

Best regards,

Saad Jbabdi

Associate Editor

PLOS Computational Biology

Kim Blackwell

Deputy Editor

PLOS Computational Biology

---

## [Editor Report · Acceptance letter]

13 Oct 2020

PCOMPBIOL-D-20-01134R1 

Model-based decoupling of evoked and spontaneous neural activity in calcium imaging data

Dear Dr Goodhill,

I am pleased to inform you that your manuscript has been formally accepted for publication in PLOS Computational Biology. Your manuscript is now with our production department and you will be notified of the publication date in due course.

With kind regards,

Laura Mallard
